# Preclinical Evaluation of the Safety, Toxicity and Efficacy of Genetically Modified Wharton’s Jelly Mesenchymal Stem/Stromal Cells Expressing the Antimicrobial Peptide SE-33

**DOI:** 10.3390/cells14050341

**Published:** 2025-02-26

**Authors:** Vagif Ali oglu Gasanov, Dmitry Alexandrovich Kashirskikh, Victoria Alexandrovna Khotina, Daria Mikhailovna Kuzmina, Sofya Yurievna Nikitochkina, Irina Vasilievna Mukhina, Ekaterina Andreevna Vorotelyak, Andrey Valentinovich Vasiliev

**Affiliations:** 1Koltzov Institute of Developmental Biology of Russian Academy of Sciences, Moscow 119334, Russia; dim.kashirsckih@gmail.com (D.A.K.); vorotelyak@yandex.ru (E.A.V.);; 2Department of Normal Physiology, Privolzhsky Research Medical University of Ministry of Health of the Russian Federation, Nizhny Novgorod 603005, Russia; dariak2294@gmail.com (D.M.K.); mukhinaiv@mail.ru (I.V.M.); 3Department of Cell Biology, Biological Faculty, Lomonosov Moscow State University, Moscow 119234, Russia

**Keywords:** mesenchymal stromal cells, antimicrobial peptide, genetic engineering, biosafety studies, cell-based therapy

## Abstract

Mesenchymal stem/stromal cells (MSCs) offer promising therapeutic potential in cell-based therapies for various diseases. However, the safety of genetically modified MSCs remains poorly understood. This study aimed to evaluate the general toxicity and safety of Wharton’s Jelly-Derived MSCs (WJ-MSCs) engineered to express the antimicrobial peptide SE-33 in an animal model. Genetically modified WJ-MSCs expressing SE-33 were administered to C57BL/6 mice at both therapeutic and excessive doses, either once or repeatedly. Animal monitoring included mortality, clinical signs, and behavioral observations. The toxicity assessment involved histopathological, hematological, and biochemical analyses of major organs and tissues, while immunotoxicity and immunogenicity were examined through humoral and cellular immune responses, macrophage phagocytic activity, and lymphocyte blast transformation. Antimicrobial efficacy was evaluated in a *Staphylococcus aureus*-induced pneumonia model by monitoring animal mortality and assessing bacterial load and inflammatory processes in the lungs. Mice receiving genetically modified WJ-MSCs exhibited no acute or chronic toxicity, behavioral abnormalities, or pathological changes, regardless of the dose or administration frequency. No significant immunotoxicity or alterations in immune responses were observed, and there were no notable changes in hematological or biochemical serum parameters. Infected animals treated with WJ-MSC-SE33 showed a significant reduction in bacterial load and lung inflammation and improved survival compared to control groups, demonstrating efficacy over native WJ-MSCs. Our findings suggest that WJ-MSCs expressing SE-33 are well tolerated, displaying a favorable safety profile comparable to native WJ-MSCs and potent antimicrobial activity, significantly reducing bacterial load, inflammation, and mortality in an *S. aureus* pneumonia model. These data support the safety profile of WJ-MSCs expressing SE-33 as a promising candidate for cell-based therapies for bacterial infections, particularly those complicated by antibiotic resistance.

## 1. Introduction

Mesenchymal stem/stromal cells (MSCs) are multipotent cells that can be found in nearly all tissues of the human body. They are characterized by the ability to differentiate into various cell types, including adipocytes, osteocytes, chondrocytes, hepatocytes, neurons, myocytes, and epithelial cells, among others [1]. Beyond their differentiation potential, numerous studies have demonstrated that MSCs possess significant immunomodulatory, anti-inflammatory properties and trophic effects [2]. In vitro, they exhibit a robust proliferative capacity while maintaining their undifferentiated multipotent state [3]. In addition, the absence of major histocompatibility complex (MHC) class I and costimulatory molecules such as CD40, CD80, and CD86, along with a minimal expression of MHC class II molecules (with the exception of minor histocompatibility antigens), contributes to the low immunogenicity of MSCs and allows them to be considered immunoprivileged [4]. These properties make MSCs an ideal candidate for the development of drugs for regenerative medicine and therapeutic applications for the treatment of a wide range of diseases.

There are two major groups of MSCs: MSCs derived from adult tissues (adult MSCs) and MSCs derived from birth-associated tissues [5]. Adult MSCs can be isolated from sources such as bone marrow (BM), adipose tissue, the lungs, heart, and peripheral blood. In contrast, MSCs derived from birth-associated tissues include cells derived from the placenta, amniotic fluid, umbilical cord (UC), umbilical cord blood (UCB), and Wharton’s Jelly (WJ) [6]. A significant advantage of birth-associated tissue-derived MSCs is their availability, which avoids additional invasive procedures and mitigates potential ethical concerns. Furthermore, birth-associated tissue-derived MSCs may exhibit enhanced properties compared to adult MSCs, such as an increased proliferative capacity, extended cellular lifespan, and greater differentiation potential [7].

Umbilical cord-derived MSCs (UC-MSCs) and Wharton’s Jelly-Derived MSCs (WJ-MSCs) have garnered considerable interest as promising platforms for the development of advanced therapy medicinal products (ATMPs) for cell-based therapies, including regenerative medicine and immunomodulatory therapy [8,9]. These cells possess several advantages over other types of MSCs, including a higher self-renewal rate and distinct immunomodulatory properties, which makes them attractive for a wide range of clinical applications. In addition, studies have demonstrated the low tumorigenic potential and capacity of these types of MSCs with the greater suppression of tumor cell growth and metastasis, making them promising candidates for therapeutic drug development [8]. In particular, UC-MSCs and WJ-MSCs have been utilized in several clinical studies and applications for the treatment of neurological (e.g., cerebral palsy), cardiovascular, liver, and kidney diseases, osteoarthritis, and immune-related and inflammatory diseases [1,9,10,11,12,13,14].

Additionally, MSCs, including UC-MSCs and WJ-MSCs, due to their antimicrobial properties, which are mediated through both direct bactericidal activity and the modulation of host innate and adaptive immune cells, can also be successfully used for treating bacterial infections and associated complications, including complicated sepsis, acute respiratory infections and pneumonia [15,16,17,18,19,20,21,22]. These effects of MSCs are particularly important to consider given the growing problem of pathogenic bacteria developing resistance to traditional antibiotics and drugs. Of particular concern is the rapid emergence of antimicrobial resistance (AMR) in respiratory pathogens such as *Staphylococcus aureus*, *Klebsiella pneumoniae*, *Pseudomonas aeruginosa*, and *Streptococcus pneumoniae* [23,24,25]. Consequently, there is an urgent need to develop innovative therapeutic strategies against pathogens with AMR, including MSC-based therapies [26,27].

The antimicrobial effects of MSCs may be attributed to their ability to produce various bioactive molecules, including a variety of antimicrobial peptides (AMPs) such as cathelicidin LL-37, hepcidin, human β-defensin-2 (hBD-2), and lipocalin-2 (Lcn2) [28,29]. For example, UC-MSCs derived from canine umbilical cord tissue have been shown to naturally express mRNAs for several AMPs, including the -X-C motif chemokine ligand 8 (*CXCL8*), Elafin (*PI3*), hepcidin (*HAMP*), Lcn2 (*LCN2*) and the secretory leukocyte protease inhibitor (*SLPI*) [30]. Moreover, BM-MSCs and UC-MSCs, as well as their conditioned medium, have been shown to exhibit antimicrobial activity and inhibit the proliferation of such pathogens as *Escherichia coli*, *P. aeruginosa*, *S. aureus*, and multidrug-resistant *K. pneumoniae* [29,31]. Preclinical and clinical trials investigating the use of UC-MSC- and WJ-MSC-based therapies for acute respiratory distress syndrome (ARDS) and pneumonia associated with viral infections of the lungs (e.g., COVID-19) are currently underway [32]. A recent meta-analysis revealed that MSC-based therapies for COVID-19-induced ARDS not only avoid adverse effects but also reduce patient mortality by 35% [33]. In addition, UC-MSC-based therapy may facilitate the resolution of pneumonia caused by antibiotic-resistant bacteria, such as *K. pneumoniae*, without exerting negative effects on physiological parameters of the body by accelerating epithelial healing, enhancing macrophage phagocytosis and improving the resolution of structural lung damage [34]. Additionally, a recent meta-analysis of studies examining the effect of UC-MSCs on the development of sepsis in animal models further supports the safety and efficacy of UC-MSCs in clinical practice [35].

Over the past decade, there has been increasing interest in enhancing the therapeutic and innate properties of MSCs, such as survival and migration, through genetic modifications using viral vectors or nonviral transfection methods [36]. Genetic engineering allows MSCs to express a variety of proteins and molecules with therapeutic potential, including chemokines, cytokines, transcription, and growth factors, without significant disruption in their differentiation potential and other vital cellular functions [37]. Importantly, genetically modified MSCs have demonstrated stronger therapeutic effects compared to native cells [38]. Several cell therapy studies have investigated the use of genetically modified MSCs in animal models of critical illness, such as sepsis, bacterial pneumonia, ARDS and lung injury, acute myocardial infarction, acute kidney injury, liver failure, Parkinson’s disease, and traumatic brain injury [37,39,40,41,42]. As such, ensuring the safety of genetically engineered MSCs is a key focus of modern studies.

The introduction of genes encoding AMPs into MSCs can significantly expand the therapeutic potential of these cells, particularly in the context of infectious diseases and inflammatory disorders. For example, the administration of distal airway stem cells genetically modified to express the native LL-37 was found to significantly enhance the recovery of injured lungs and protect animals from bacterial pneumonia and hypoxemia [41]. Similarly, a recent study demonstrated that the genetic modification of human UC-MSCs to express the antibacterial fusion peptide containing BPI21 and LL-37 significantly enhances their antibacterial and toxin-neutralizing activities, leading to improved survival and reduced organ damage in a sepsis mouse model, thus highlighting the potential of engineered MSCs as advanced cell-based therapies for severe bacterial infections [43]. However, it should be noted that the LL-37 application is associated with certain limitations [44,45,46,47]. High doses of cathelicidin LL-37, as well as its peptide fragments, which are generated due to the high susceptibility of LL-37 to proteolytic degradation, exhibit significant toxicity toward human cells [47,48]. This can lead to adverse outcomes and contribute to the development of chronic inflammatory diseases (e.g., atopic dermatitis, rosacea, psoriasis, and hidradenitis suppurativa) [44]. Additionally, there is evidence suggesting that certain pathogens are able to develop adaptive resistance mechanisms to LL-37 [45,46,47]. In this regard, the development of novel AMPs with targeted immunomodulatory effects, alongside direct antimicrobial activity, represents a promising avenue in the search for new therapeutic strategies.

To date, numerous derivatives of LL-37 or its analogs have been developed and synthesized [47,49,50]. One such peptide is SE-33, a retrosequence of the C-terminal part of LL-37. This peptide is characterized by significantly reduced toxicity while maintaining high antibacterial and antifungal activity against *Candida albicans*, *Cryptococcus neoformans*, *Rhodotorula mucilaginosa*, *Trichosporon cutaneum*, and *Geotrichum* species [51,52,53]. Similarly to natural cathelicidin LL-37, SE-33 forms an amphipathic alpha-helix but with an inverted amino acid sequence. Despite this structural inversion, SE-33 retains physicochemical properties comparable to LL-37. In vitro studies have demonstrated that SE-33 exhibits bactericidal activity against *S. aureus* and *E. coli*, comparable to LL-37 [51,53]. Furthermore, in vivo experiments have shown that SE-33 effectively reduces the pathogen load without inducing side effects or pathological changes in mice. These results indicate that SE-33 possesses potent antibacterial activity while maintaining a favorable safety profile, supporting its potential as a promising antimicrobial agent.

Taking into account the above, the genetic modification of MSCs to enhance the secretion of natural AMPs or their analogs opens up new possibilities for the development of advanced cell-based therapies for pulmonary diseases, both caused and complicated by bacterial infections. Despite the encouraging results of clinical trials, limitations, and unresolved questions remain concerning the long-term safety and use of MCSs in widespread clinical practice [35]. Some evidence indicates an increased risk of thromboembolic events in patients with chronic kidney disease, as well as following transplantation when treated with UC-MSCs [54,55]. Furthermore, as with all genetically modified cells, there are concerns regarding their long-term safety. Potential risks include unpredictable immune responses, mutagenesis, and effects on the microbiome. These issues highlight the need for further study of the safety of cell-based therapies. Although there are some data regarding the safety of MSCs for therapeutic purposes, information on genetically modified MSCs remains limited, raising concerns about their clinical application.

We hypothesize that the genetic modification of WJ-MSCs to express SE-33 enhances their antimicrobial activity while maintaining a safety profile comparable to native MSCs. This could offer a novel and potent cell-based therapeutic strategy for the treatment of bacterial infections, including those complicated by antibiotic resistance. Additionally, we propose that the antimicrobial effect of SE-33 and the immunomodulatory properties of WJ-MSCs synergistically enhance pathogen clearance and more effectively mitigate inflammatory damage compared to native WJ-MSCs.

Therefore, the present study aims to evaluate the general toxicity and safety profile of genetically modified WJ-MSCs expressing the antimicrobial peptide SE-33 (WJ-MSC-SE33) following single and repeated administration to mice, with a particular focus on immunotoxicity and immunogenicity. These assessments are essential for advancing the clinical application of WJ-MSC-SE33 as a potential therapeutic agent. Unlike native WJ-MSCs, genetically modified WJ-MSC-SE33 is engineered to enhance antimicrobial activity while reducing toxicity. This modification is hypothesized to confer enhanced antibacterial and immunomodulatory properties, making WJ-MSC-SE33 a promising candidate for treating bacterial infections, particularly those complicated by antibiotic resistance. Furthermore, this study aims to investigate the antimicrobial activity of WJ-MSC-SE33 in vivo using a bacterial lung infection model induced by *S. aureus*. Specifically, we evaluate the impact of WJ-MSC-SE33 on survival rates, bacterial load reduction, and the attenuation of inflammatory responses in the lungs.

## 2. Materials and Methods

### 2.1. Plasmid Vector Construction

The plasmid vector pVIDBse33 was constructed according to a previously developed protocol [51,52]. Briefly, the SE-33 peptide-encoding DNA fragment was inserted into the pcDNA3.1(+) vector under the control of the cytomegalovirus (CMV) IE promoter (Figure 1). This plasmid also contains a kanamycin/neomycin resistance gene, SV40 polyadenylation (polyA) fragment, HSV-tk poly A fragment, SV40, pUC origin of replication (Ori), f1 Ori, and SV40 Ori. The pVIDBse33 plasmid was amplified in *E. coli* XL1-Blue cells grown in an LB medium (Carl Roth, Karlsruhe, Germany) supplemented with 25 µg/mL kanamycin (AppliChem, Darmstadt, Germany) to an optical density of OD_600_ = 0.6. The plasmid was then purified using a QIAGEN Plasmid Mini Kit (Qiagen, Hilden, Germany) according to the manufacturer’s protocol. Upon the transfection of MSCs, this plasmid expresses the SE-33 peptide with the amino acid sequence SETRPVLNRLFDKIRQVRKEFGKIKEKSRKFM.

### 2.2. Mesenchymal Stem/Stromal Cell Culture and Transfection

Human Wharton’s Jelly-Derived mesenchymal stem/stromal cells (WJ-MSCs) were obtained from the Cell Culture Collections for Biotechnological and Biomedical Research at the Koltzov Institute of Developmental Biology of Russian Academy of Sciences (Moscow, Russia). The isolation, characterization, identification, trilineage differentiation, and storage of WJ-MSCs were then documented in the cell passport, available at https://www.en.idbras.ru/index.php/en/institute/bio-resources/88-cell-culture-collection (accessed on 15 December 2024). The cells used in this study are designated as WJMSC (20200528). The cells used were derived from a single donor and remained viable up to passage 18. The cells were not immortalized.

The WJ-MSCs were thawed from cryo-storage, washed to remove the cryopreservation medium, and resuspended in a complete growth medium. The cells were cultured in Dulbecco’s Modified Eagle Medium/Nutrient Mixture F-12 (DMEM/F-12) (PanEco, Moscow, Russia) supplemented with 10% fetal bovine serum (FBS) (Capricorn Scientific, Ebsdorfergrund, Germany), 2 mM glutamine, 4.5 g/L glucose, 100 U/mL penicillin-streptomycin (PanEco, Moscow, Russia), and 10 mM sodium pyruvate at 37 °C in a humidified atmosphere containing 5% CO_2_. The WJ-MSCs (passages 3–4) were seeded at a density of 1 × 10^5^ cells per well in a 12-well plate and allowed to adhere to the plate surface for 24 h prior to transfection. Then, the cells were transfected in OptiMEM medium (Gibco, Thermo Scientific, Waltham, MA, USA) using 1 μg of plasmid DNA and the Lipofectamine LTX PLUS reagent (Thermo Scientific, Waltham, MA, USA) in accordance with the manufacturer’s protocols. After 72 h, transfection was stopped by washing the cells twice with sterile phosphate-buffered saline (PBS) (PanEco, Moscow, Russia). The transfected WJ-MSCs were then harvested using 0.25% trypsin-EDTA (PanEco, Moscow, Russia) at 37 °C, neutralized with Versene solution (PanEco, Moscow, Russia), washed twice with PBS, and transferred into a complete growth medium without penicillin-streptomycin for an additional 24 h of recovery. The cells were then harvested and cryopreserved at a density of 1 × 10^5^ cells/mL until further use. Aliquots of the cells and culture medium (supernatants) were collected and stored at −79 °C for the antibacterial activity assessment and high-performance liquid chromatography (HPLC). For animal studies, the frozen cells were thawed, cultured for no more than 48 h (passages 5–6), harvested, and resuspended in 0.9% NaCl at the required concentration, depending on the administered dose.

### 2.3. Antimicrobial Activity Assessment

Before each experiment, *E. coli* were seeded from frozen stocks and grown overnight at 37 °C in liquid LB Broth (Luria–Bertani, Miller) (HiMedia, Mumbai, India) with slight agitation. The bacterial cells were washed once and resuspended in PBS, and the optical density (OD_600_) of the suspension was measured. The antimicrobial activity of native and genetically modified WJ-MSCs was evaluated using the agar well diffusion method. Briefly, sterile 2% agar, cooled to 50 °C, was inoculated with the suspension of *E. coli* (to a final concentration of 1.45 × 10^7^ CFU/mL). The agar–bacteria mixture was poured into sterile Petri dishes, and wells were formed in the agar. Once solidified, suspensions of the analyzed WJ-MSCs or supernatants were introduced into the wells. Ampicillin (25 µg/µL) and SE-33 peptide (1 mg/mL) solutions were used as controls. Plates were incubated at 37 °C for 24 h, after which the diameter of the inhibition zones was measured. Only WJ-MSC-SE33 cells exhibiting antimicrobial activity were selected for further studies.

### 2.4. High-Performance Liquid Chromatography

HPLC was used to evaluate the levels of the antimicrobial peptide SE-33 in cells and culture media. Suspensions of native WJ-MSC and WJ-MSC-SE33 cells (1 × 10^5^ cells) were centrifuged at 2000× *g*, and the resulting pellets were resuspended in 3 mL of 20 mM Tris-HCl (pH 6.8) in 8 M urea solution. For culture media analysis, 100 mL of supernatants were used. Samples were applied to a 1 mL HiTrap SP Sepharose FF column pre-equilibrated with 20 mM Tris-HCl (pH 6.8) in 8 M urea solution. Columns were washed with 10 mL of 20 mM Tris-HCl (pH 6.8), and the target product was eluted with 3 mL of a solution containing 75 mM of NaCl and 20 mM of TrisHCl (pH 6.8). The eluates were diluted 5-fold with 0.1% trichloroacetic acid (TCA) and applied to a Chromabond C18 Hydra column (6 mL/500 mg), pre-equilibrated with 0.1% TCA. After washing the columns with a solution of 0.1% TCA in 5% acetonitrile, the product was eluted with 6 mL of 90% acetonitrile. The eluates were evaporated under vacuum using a Savant SC-210A vacuum centrifuge (Savant, Thermo Scientific, Waltham, MA, USA), and the dry residues were dissolved in 100 μL of a solution of 0.1% TCA and 5% acetonitrile. Chromatographic analysis was performed on a Breeze QS HPLC system (Waters, Milford, KT, USA) using an Ambercrome 100-5-C18 column (4.6 × 250 mm) with a linear acetonitrile gradient. The SE-33 peptide concentration in the samples was determined by the peak area using a calibration curve constructed on the basis of standard solutions of the SE-33 peptide at concentrations ranging from 0.2 to 1.2 μg/mL.

### 2.5. Animal Studies

The studies were conducted on sexually mature male and female C57BL/6 mice with an average body weight of 20 g and an age of 8 weeks. The animals were sourced from the “Andreevka”, branch of the Scientific Center of Biomedical Technologies of the Russian Federal Medical and Biological Agency (Russia). Mice underwent a 14-day acclimatization period. The animals were housed in individually ventilated polysulfone cages ISO RAIR 4x8 (LabProducts, Aberdeen, WA, USA) with Rehofix MK2000 corn cob bedding (J. Rettenmaier & Söhne GmbH, Rosenberg, Germany). The animal housing environment followed a 12 h light/dark cycle with ad libitum access to water and standard laboratory feed. The group assignment was based on body weight, ensuring no more than a 10% deviation from the average group weight. All experimental procedures were approved by the Local Ethics Committee of the Privolzhsky Research Medical University of the Ministry of Health of the Russian Federation (Protocol No. 06, dated 29 April 2022) and adhered to the guidelines for the care and use of laboratory animals [56].

### 2.6. Administration of Genetically Modified WJ-MSCs to Animals

Immediately prior to administration, WJ-MSC-SE33 cells were resuspended in saline solution (0.9% NaCl). Mice fasted for 8 h prior to intravenous injection. The injection volume was calculated based on body weight, with doses of 1 × 10^5^, 2.5 × 10^5^, or 5 × 10^5^ WJ-MSC-SE33 cells per mouse (equivalent to 0.5 × 10^7^, 1.25 × 10^7^, or 2.5 × 10^7^ cells/kg for a 20 g mouse). The course of repeated administration was carried out over two weeks with three injections per week. The maximum achievable dose of 0.2 mL/mouse (2.5 × 10^7^ cells/kg) was chosen as the maximum administered dose. The maximum dose administered intravenously (5 × 10^5^ cells/mouse in 0.2 mL) was selected to avoid adverse physiological effects, such as thrombus formation and reflex reactions, associated with excessive injection volumes [57].

To assess general toxicity after a single administration, 120 mice were divided into five groups (n = 12 males and 12 females per group). WJ-MSC-SE33 cells were administered intravenously to three experimental groups, while the control groups received either 0.9% NaCl or a suspension of native MSCs (2.5 × 10^7^ cells/kg). Mice were observed for 14 days post-injection and euthanized on day 15 by CO_2_ asphyxiation (Figure 2A).

For the repeated administration of toxicity studies, 240 mice were divided into five groups (n = 24 males and 24 females per group). WJ-MSC-SE33 cells were administered intravenously three times per week for two weeks to three experimental groups, while the control groups received either 0.9% NaCl or a suspension of native MSCs (2.5 × 10^7^ cells/kg). Mice were observed for 14 days post-injection and euthanized on days 15 and 29 by CO_2_ asphyxiation (Figure 2B).

### 2.7. Mortality and Clinical Signs

Mice were observed for mortality and signs of toxicity for the first 4 h post-injection (30 min, 1 h, 2 h, 4 h intervals) and daily thereafter for 14 days. In the case of repeated administrations, animals were monitored daily during the 14-day treatment period and for 14 days post-treatment. Any cases of death that occurred during the study period were recorded.

A clinical examination of the general condition and behavioral responses was conducted daily through a visual assessment of the following parameters: (1) in-cage activity (tremor, response to tactile, pain, auditory, and visual stimuli); (2) emotional state parameters (in response to environmental changes, including transfer reactions, responses to handling, evidence of urination and defecation, and vocalizations during handling); (3) stress markers and reflexes (barbering behaviors, and simple reflex assessments, including startle, righting, corneal, and auricular reflexes); (4) ocular health (measurement of eye-slit width for signs of swelling, lacrimation, or conjunctival hyperemia); (5) nasal and ear conditions (the color and temperature assessment of the nose and ears, checking for pathological discharges, contaminants, dental health, and presence of salivation); (6) respiratory patterns (frequency, amplitude, and rhythm); (7) musculoskeletal health (motor activity, muscle tone, and any vocalizations indicating pain); (8) dermal condition (inspection of skin and fur quality, including visible mucous membranes); and (9) nutritional intake (water and food consumption). Any clinical signs observed, including their onset, progression, or resolution, were documented.

### 2.8. Body Weight and Temperature

In the case of a single administration of WJ-MSC-SE33, body weight, and rectal temperature were measured one day before treatment and on days 7 and 14 post-treatment. For repeated administrations, these parameters were assessed one day before treatment, on day 14, and on day 28. Body weight was measured with a CAS AD-05H electronic scale (CAS Corp, Seoul, Republic of Korea), and rectal temperature was recorded using an Omron Eco Temp Smart (MC-341-RU) thermometer (Omron, Kyoto, Japan).

### 2.9. Necropsy and Histopathological Studies

A pathological examination was conducted on day 15 for single administration groups and on days 15 and 29 for repeated administration groups. Animals were sacrificed by CO_2_ asphyxiation and subjected to necropsy. External examination was performed to identify any gross pathological signs, followed by macroscopic inspection of removed organs. The gross pathological changes were documented by examination under a microscope. All observed pathological changes were documented, and organ masses were recorded using an AND DX-300 scale (A&D, Tokyo, Japan). The organ mass coefficient (*MC*) was calculated as follows:(1)MC=(mM)×1000
where m is the organ mass (g), and M is the body mass of the animal (g).

For histopathological examination, tissues were fixed in a 10% neutral buffered formalin, dehydrated in a graded ethanol series, and embedded in paraffin. Paraffin Sections (5–7 μm) were cut using an SM 2000R microtome (Leica, Wetzlar, Germany), stained with hematoxylin and eosin, and examined under a DM1000 light microscope (Leica, Wetzlar, Germany). Digital images were captured with a DFC290 camera (Leica, Wetzlar, Germany).

A semi-quantitative scoring system was used for lung histopathological studies based on the following parameters:(1)Infiltration or aggregation of inflammatory cells in the airspace or vessel wall: 0—none; 1—wall only; 2—a few cells (1–5) in the airspace; 3—intermediate; 4—severe (airspace is congested).(2)Interstitial congestion and hyaline membrane formation: 1—normal lung; 2—moderate (<25% of the lung area); 3—intermediate (25–50% of the lung area); 4—severe (>50% of the lung area).(3)Hemorrhage: 0—absent; 1—present.

### 2.10. Functional Assessment of the Central Nervous System

The CNS function was evaluated before administration and on day 14 post-administration for single administration groups, as well as on days 14 and 28 for repeated administration groups. CNS activity was assessed using the open field test, where motor activity, exploratory behavior, and emotional status were evaluated. Behavioral data were recorded using Open Field apparatuses (PanLab, Barcelona, Spain; Stoelting, Wood Dale, IL, USA) and analyzed via proprietary software from PanLab (Barcelona, Spain) and Stoelting (Wood Dale, IL, USA), counting the number of specific behaviors over a 5 min interval.

The following behaviors were measured: (1) vertical motor activity (the number of rearing events to assess exploratory behavior); (2) horizontal motor activity (the time spent moving to assess general activity). Emotional status was evaluated by the duration (s) of (1) the freezing reaction; (2) grooming behavior; and (3) sniffing behavior.

### 2.11. Hematological and Biochemical Blood Analysis

Hematological parameters (erythrocytes, leukocytes, and platelet counts; hemoglobin concentration; leukocyte differential count; and ESR) were assessed on day 15 for single administration groups and on days 15 and 29 for repeated administration groups. Blood parameters were measured with a Mindray BC-3200 hematology analyzer (Mindray, China). The leukocyte differential count was performed on Romanovsky-stained blood smears, examined under a DM1000 microscope (Leica, Germany) using an immersion objective (10× eyepiece, 90× objective). ESR was determined using the method of T.P. Panchenkov.

Biochemical serum analysis was conducted with a BS-120 biochemical analyzer (Mindray, Shenzhen, China), assessing parameters such as total protein, albumin, creatinine, urea, glucose, cholesterol, triglycerides, total bilirubin, alkaline phosphatase, aspartate, alanine aminotransferase activity, and levels of calcium, potassium, and sodium.

### 2.12. Renal Function Assessment

Renal functional status was evaluated on day 15 for single administration groups and on days 15 and 29 for repeated administration groups through urinalysis. For urine collection, mice were housed in individual metabolic cages for 4 h. Urinalysis included the measurement of specific gravity and pH using a LAURA SMART urine analyzer (Erba Lachema, Brno, Czech Republic) with DekaPHAN LAURA test strips (Erba Lachema, Brno, Czech Republic). Urine was analyzed for protein, glucose, ketones, and bilirubin. Sediment microscopy was conducted with a DM1000 light microscope (Leica, Wetzlar, Germany), counting leukocytes and erythrocytes (8× eyepiece, 10× and 40× objectives).

### 2.13. Assessment of Local Tolerance

Local tolerance was evaluated alongside toxicity studies using repeated administration groups of mice (n = 240), *incorporating* a histological examination to assess signs of local inflammatory responses in the tail vein wall and surrounding tissues resulting from WJ-MSC-SE33 administration.

### 2.14. Assessment of Systemic Anaphylactic Response

To assess the sensitization potential of genetically modified WJ-MSCs, a systemic anaphylaxis study was conducted in mice (n = 96) divided into 8 groups (n = 6 males and 6 females per group). Four groups received endolaryngeal, and four received intravenous administration. Experimental groups received WJ-MSC-SE33, while control groups received either 0.9% NaCl (negative control) or 0.6% hen egg-white (HEW) solution (positive control, at 1 mg/300 g body weight). Sensitization in the experimental groups involved three administrations of WJ-MSC-SE33 at one-day intervals via either endolaryngeal or intravenous routes. The positive control group was sensitized intragastrically with HEW over three days, with ovalbumin as the allergen.

On day 14, a resolving dose of WJ-MSC-SE33 (equivalent to the sensitizing dose) was administered intracardially to the experimental and negative control groups. Ovalbumin was administered intracardially to the positive control group at 1 mg/300 g body weight. The anaphylactic response intensity was calculated using the Weigle reaction index (*I_w_*) [58]:(2)$$I_{w}=\frac{\left(N\times 4)+(N_{1}\times 3)+(N_{2}\times 2)+(N_{3}\times 1)+(N_{4}\times 0\right)}{N+N_{1}+N_{2}+N_{3}+N_{4}}$$
where N is the count of animals that died; N_1_ indicates animals with severe shock; N_2_ indicates moderate shock; N_3_ is a mild shock; and N_4_ is without shock. Full mortality in the group results had an *I_w_* of 4, with values of 3, 2, and 1 for severe, moderate, and mild shock, respectively.

### 2.15. Immunotoxicity and Immunogenicity Studies

#### 2.15.1. Humoral Immune Response

The humoral immune response was evaluated via a hemagglutination assay to assess the degree and duration of antibody production following the repeated intravenous administration of WJ-MSC-SE33. Animals (n = 40, 4 groups of 10 males each) received WJ-MSC-SE33 at doses of 0.5 × 10^7^ or 2.5 × 10^7^ cells/kg, followed by intraperitoneal injection of a suboptimal dose of washed sheep erythrocytes (5 × 10^7^ cells/mouse) one-hour post-administration. Two control groups received either a 0.9% NaCl solution or a native WJ-MSC suspension (2.5 × 10^7^ cells/kg). On day 7, antibody titers were measured in blood serum using direct hemagglutination [59]. Serum titer was defined as the last positive dilution showing hemagglutination.

#### 2.15.2. Cellular Immune Response

The cellular immune response was assessed through a delayed-type hypersensitivity (DTH) reaction using trinitrobenzenesulfonic acid (hapten) as the antigen. Mice (n = 40, 4 groups of 10 males each) received WJ-MSC-SE33 at doses of 0.5 × 10^7^ and 2.5 × 10^7^ cells/kg intravenously three times weekly for one week. Animals of the control groups were intravenously injected with 9% NaCl solution in similar volumes. For DTH induction, the antigen was administered twice: once for sensitization and again five days later for resolution. A 0.9% NaCl solution was injected into the control animal paw. The inflammatory response (edema formation) was measured 24 h after antigen injection by oncometric measurements of hind paw volume using a plethysmometer (Open Science, Krasnogorsk, Russia).

The reaction index (*I_r_*, %) was calculated as follows:(3)$$I_{r}=(\frac{V_{t}-V_{k}}{V_{k}})\times 100\%$$
where V_t_ is the antigen-injected paw volume, and V_k_ is the control paw volume.

#### 2.15.3. Phagocytic Activity Assessment

Phagocytic activity was determined by the ability of peritoneal macrophages to engulf colloidal ink particles. Animals (n = 40) were divided into four groups (n = 10 males per group). Animals in experimental groups received WJ-MSC-SE33 intravenously at doses of 0.5 × 10^7^ and 2.5 × 10^7^ cells/kg intravenously, while control animals received a 0.9% NaCl solution intravenously in corresponding volumes. Macrophages were isolated from the peritoneal cavity 24 h post-administration. The phagocytic activity of peritoneal macrophages was determined by their ability to engulf ink particles that were introduced intraperitoneally as a 0.05% suspension in a 2 mL volume.

#### 2.15.4. Lymphocyte Blast Transformation Assessment

Spleen lymphocytes (splenocytes) undergoing in vitro proliferation after the administration of WJ-MSC-SE33 were assessed using B-cell mitogen (lipopolysaccharide (LPS) from *E. coli* O111:B4). Animals (n = 40) were divided into 4 groups (n = 10 males each). Mice from the experimental groups received WJ-MSC-SE33 cells at 0.5 × 10^7^ and 2.5 × 10^7^ cells/kg intravenously three times weekly for one week, while control animals were administered 0.9% NaCl intravenously in corresponding volumes. After euthanasia, spleens were harvested, and splenocyte suspensions were prepared, seeded in a 96-well plate at a concentration of 2 × 10^6^ cells/mL in RPMI-1640 medium, supplemented with 10% FBS, 10 mM Hepes, 2 mM L-glutamine, 10 mM 2-mercaptoethanol, 100 mg/mL streptomycin, and 100 units/mL penicillin, and incubated with 20 µg/mL LPS at 37 °C with 5% CO_2_ for 96 h. Then, the blast cell percentage was determined and analyzed.

#### 2.15.5. Immunogenicity Studies of WJ-MSC-SE33

Immunogenicity was evaluated by measuring specific immunoglobulin (IgM and IgG) titers after the repeated administration of WJ-MSC-SE33. Mice (n = 240, 8 groups of 30) received WJ-MSC-SE33 endolaryngeally or intravenously at doses of 0.5 × 10^7^ and 2.5 × 10^7^ cells/kg. Control groups received 0.9% NaCl or native WJ-MSCs (2.5 × 10^7^ cells/kg). Blood was collected during the 14th, 21st, and 28th days post-treatment, and IgM and IgG levels were quantified by the enzyme-linked immunosorbent assay (ELISA) using the MEA543 Mu and HEA544 Mu kits (Cloud-Clone Corp, Wuhan, China) with an Epoch Microplate Reader (BioTek, Winooski, VT, USA), following the manufacturer’s protocols.

### 2.16. Induction of Infectious Lung Disease and Sample Collection

Bacterial pneumonia was chosen as the experimental model of infectious lung disease. Mice were intranasally infected with a suspension of *S. aureus* (clinical isolate strain 1456) at a dose of a 1.1 × 10^9^ colony-forming unit (CFU) in 50 μL phosphate-buffered saline (Figure 3). Lung colonization was subsequently quantified.

To assess the therapeutic efficacy against bacterial lung infection, animals were divided into five groups (n = 20 each). Mice were intravenously administered cell suspensions on days 1, 3, and 5 at doses of 1 × 10^5^ cells/mouse (0.5 × 10^7^ cells/kg), 2 × 10^5^ cells/mouse (1 × 10^7^ cells/kg), and 3 × 10^5^ cells/mouse (1.5 × 10^7^ cells/kg) 24 h post-infection with *S. aureus* (Figure 3). Control groups received either isotonic sodium chloride solution (0.9% NaCl) or native WJ-MSCs (0.5 × 10^7^ cells/kg).

Bronchoalveolar lavage fluid (BALF) was obtained on days 5 and 8 (n = 10/group/day) (Figure 3). BALF samples were collected to evaluate the bacterial load, protein content, and leukocyte infiltration. Mortality rates were recorded over 8 days.

### 2.17. CFU Quantification in BALF

BALF (0.1 mL) was diluted in 0.9 mL of saline and plated (0.5 mL) on nutrient agar containing N-cetylpyridinium chloride monohydrate (CPC agar). Plates were incubated at 37 °C for 48 h, and CFUs were enumerated.

### 2.18. Protein Quantification in BALF (Lowry Method)

The protein concentration was determined by adding 2 mL of the working reagent (2% Na_2_CO_3_ in 0.1 N NaOH; 0.5% CuSO_4_·5H_2_O in 1% sodium citrate) to 0.4 mL of BALF. Samples were incubated at 25 °C for 10 min, followed by the addition of a 0.2 mL Folin–Ciocalteu reagent and a further 30 min incubation in the dark. Absorbance was measured at 740 nm using a Synergy HTX spectrophotometer (BioTek, Winooski, VT, USA).

### 2.19. Leukocyte Quantification in BALF

BALF was centrifuged (1500 rpm, 8 min), and erythrocytes were removed using ammonium buffer (pH 7.2–7.4). Cell pellets were resuspended in 0.1 mL of Hanks’ solution, and leukocytes were enumerated in a Goryaev chamber.

### 2.20. Statistical Analysis

Data were analyzed using IBM SPSS Statistics 26 software (IBM, Armand, NY, USA). Descriptive statistics included the mean (M) and standard error of the mean (SEM). The normality of data distribution in the groups was assessed using the Shapiro–Wilk test. Due to the non-normal data distribution, nonparametric tests were employed as follows: the Mann–Whitney *U*-test was used for two independent groups, and the Kruskal–Wallis test was used for multiple comparisons. Survival analysis was performed using a log-rank (Mantel–Cox) test. Differences between groups were considered statistically significant at a *p*-value of *p* < 0.05.

## 3. Results

### 3.1. Toxicological Evaluation of Genetically Modified MSCs Expressing the SE-33 Peptide in Animals Following Single and Repeated Administrations

#### 3.1.1. Evaluation of the SE-33 Peptide Level in WJ-MSC-SE33

HPLC analysis revealed that the retention time of the SE-33 peptide on the system used was 13.2–13.5 min. It was shown that the SE-33 peptide was absent both in the cell lysate and for supernatants of native WJ-MSCs (Figure 4A). At the same time, a peak with a retention time of 13.5, corresponding to the SE-33 peptide, was registered in WJ-MSC-SE33 lysates. Similarly, a peak with the same retention time was detected in supernatants from WJ-MSC-SE33 (Figure 4A). Quantitative analysis using the calibration curve (Figure 4B) showed that 1 × 10^5^ WJ-MSC-SE33 cells contained an average of 0.8 μg of the SE-33 peptide, while supernatants from WJ-MSC-SE33 contained an average of 0,75 ng/mL of the peptide. These findings suggest that the SE-33 peptide is predominantly produced and accumulated inside cells, with minimal secretion into the extracellular environment. Furthermore, an antimicrobial activity assay against *E. coli* confirmed that WJ-MSC-SE33 exhibited significantly higher antimicrobial activity compared to their supernatants, whereas native WJ-MSCs and their supernatants demonstrated a complete absence of antimicrobial activity (Figure 4C).

#### 3.1.2. Clinical Signs and Mortality Following WJ-MSC-SE33 Administration

Over a 14-day observation period following a single intravenous administration of WJ-MSC-SE33 at therapeutic and supratherapeutic doses, no mortality was recorded in male or female animals. In studies involving repeated administration, no deaths occurred during the 14-day administration phase or within the 14-day post-administration period across all administered doses of WJ-MSC-SE33. Thus, no mortality was observed across the experimental and control groups, irrespective of single or repeated intravenous administrations of genetically modified MSCs.

Clinical observations were conducted daily for 14 days following a single injection and for 28 days following the initiation of repeated administrations of WJ-MSC-SE33. No gender-specific differences were observed in either administration regimen. No significant clinical signs of intoxication were noted between the animal groups that received administrations of WJ-MSC-SE33. Across all study groups, animals exhibited no tremors, and responses to tactile, pain, sound, and light stimuli were consistent with normal behavior. Reflexes (including startle, rollover, corneal, and ear reflexes) were intact, and responses to environmental changes were within the expected norms. No urination, defecation, or vocalization occurred upon transfer between the cages. Ocular and nasal assessments showed normal findings without discharge or edema. Skin, coat, and visible mucous membranes remained unremarkable, and breathing patterns (frequency, amplitude, rhythm) and muscle tone were consistent with normal levels. Limb morphology and coat conditions remained unaffected, with no changes in motor, feeding behavior, or water and food consumption observed between the treated and control groups.

#### 3.1.3. Effects of WJ-MSC-SE33 on Body Weight and Rectal Temperature

No statistically significant differences in body weight gain were observed over a 14-day period in groups receiving a single intravenous WJ-MSC-SE33 administration compared to the control groups (Table 1). Body weight measurements were taken on day 0 (one day prior to treatment), day 7, and day 14 (one day prior to sacrifice). Rectal temperature assessments similarly showed no significant differences between experimental and control groups (Table 2). In the repeated-administration groups, no significant differences in body weight were observed between the treated and control animals (Appendix A).

Rectal temperature data for both single and repeated WJ-MSC-SE33 administration groups remained within the normal range for mice, with no significant variations detected across all dosages and between treatment and control groups throughout the study period (Table 2; Appendix A). Therefore, no significant differences were found between the WJ-MSC-SE33 and control groups for rectal temperature.

#### 3.1.4. Effects of WJ-MSC-SE33 on Central Nervous System Functional State

A single intravenous administration of WJ-MSC-SE33 did not result in significant changes in open-field test parameters between the experimental and control groups (Table 3), with no gender-related differences observed. The repeated administration of WJ-MSC-SE33 to animals at all doses tested over 14 days similarly showed no significant alterations in behavioral parameters, either immediately post-administration or after a 14-day observation period (Appendix A). Thus, single and repeated intravenous WJ-MSC-SE33 administration at all studied doses demonstrated no impact on the central nervous system (CNS) functional parameters as assessed by reflex activity in the open field test, with no gender-specific effects.

#### 3.1.5. Effects of WJ-MSC-SE33 on Hematological Parameters

Hemoglobin, erythrocyte, leukocyte, platelet counts, and the erythrocyte sedimentation rate (ESR) did not significantly differ from control values in the groups with the single administration of WJ-MSC-SE33 across all doses (Table 4). Additionally, the distribution of leukocyte subtypes in the leukocyte differential count remained unchanged (Table 5), with no gender differences observed. Following the repeated administration of WJ-MSC-SE33, no significant alterations in hemoglobin, erythrocyte, leukocyte, or platelet counts were observed compared to the controls. However, 14 days post-administration, a statistically significant reduction in leukocyte count was detected in experimental animals treated with WJ-MSC-SE33 and control animals treated with native WJ-MSCs compared to the control group of animals who received injections of 0.9% NaCl. Leukocyte distribution and ESR remained consistent across the groups throughout the study period (Appendix A). Thus, repeated injections in mice with WJ-MSC-SE33 showed no enduring hematological changes except for a transient reduction in the leukocyte count observed 14 days post-treatment.

#### 3.1.6. Effects of WJ-MSC-SE33 on Blood Biochemical Parameters

No statistically significant differences in primary serum biochemical parameters were found in single WJ-MSC-SE33 administration groups at all tested doses compared to the controls (Table 6). Repeated administration also showed no significant changes in biochemical markers over the entire observation period (Appendix A). These results indicate that the single and repeated intravenous administration of WJ-MSC-SE33 at all doses did not impact biochemical parameters.

#### 3.1.7. Effects of WJ-MSC-SE33 on Renal Function Parameters

No significant alterations in renal excretory function were observed following a single intravenous WJ-MSC-SE33 administration at all tested doses, with no gender-related differences (Table 7). Repeated administration showed similar unaltered renal function parameters in the treated versus control groups. Therefore, both single and repeated WJ-MSC-SE33 administration had no impact on the renal functional state in mice (Appendix A).

#### 3.1.8. Effects of WJ-MSC-SE33 on Organ Mass Coefficients

The single administration of WJ-MSC-SE33 at all doses did not influence organ mass coefficients in male or female mice (Table 8). Likewise, the repeated administration resulted in no detectable changes in organ mass coefficients, and no gender differences were noted.

Fourteen days after the administration of WJ-MSC-SE33 or native WJ-MSCs, organ mass coefficients in both experimental and control groups remained comparable to those of the control group that received saline injections (Appendix A). These findings indicate that repeated WJ-MSC-SE33 administration at all doses did not alter the parameter of organ mass coefficients.

#### 3.1.9. Macroscopic and Histological Analysis of Internal Organs

A comprehensive macroscopic assessment of internal organs was conducted 14 days post single intravenous injections and following repeated intravenous administrations of WJ-MSC-SE33. No observable differences were noted between the experimental groups or between the experimental and control groups (Figure 5 and Figure 6). Thus, the necropsy data described herein apply consistently across all groups. Additionally, in the absence of significant differences between WJ-MSC-SE33-treated groups and controls (injected with 0.9% NaCl or native WJ-MSC), histological data are presented as an average across all groups.

In the thoracic and abdominal cavities, no abnormalities were detected in organ position or morphology. Detailed macroscopic and histological descriptions of individual organs follow below.

Brain. The meninges appeared thin and transparent, with brain tissue of normal density and a smooth cortical surface. The gray and white matter were distinctly visible on brain frontal sections, with cerebral ventricles of typical dimensions. Cortical architecture remained intact, with no evidence of neuronal loss or signs of acute or chronic dysfunction. Neuronal nuclei displayed light, well-defined nucleoli with thin nuclear membranes. Subcortical layers contained organized bundles of neuronal fibers and clusters of neurons in the form of individual nuclei. Midbrain and medulla oblongata neurons exhibited normal nuclear morphology, with chromatin content and clear nucleoli.

Heart. The heart exhibited normal size, shape, and coloration. Both ventricles contained a small volume of dark liquid blood. The heart valves were smooth, thin, and glossy. Myocardial tissue was homogeneously cherry-brown, with distinct transverse striations. Epicardial and endocardial layers consisted of thin connective sheets with fibroblast inclusions. Coronary artery branches were thin-walled and well endothelialized, with prominent basement membranes, media, and adventitia. Cardiomyocytes were intact, with a thin sarcolemma, oxyphilic sarcoplasm, and centrally located basophilic oval nuclei containing a normal amount of chromatin. Visible myofibrillar striations were observed in all parts of the heart.

Thymus. The thymus was triangular and moderately dense, with a well-defined lobular structure. The cortex of each lobule was densely populated with lymphocytes, while the medulla contained fewer lymphocytes and clusters of epithelial cells and stratified bodies. The stromal blood content was moderate.

Lungs. The tracheal and bronchial lumens appeared unaffected, with smooth, pale mucosal linings. The lungs were pink, airy, and free of palpable consolidations, with focal areas of plethora. The sheets of the pleura, pericardium, and peritoneum were thin and smooth. Alveoli were well expanded with thin walls, while bronchi and bronchioles were lined with intact cuboidal and columnar epithelia, and their lumen was clear. The nuclei of the alveolar epithelium cells were clear, and the cytoplasm was oxyphilic.

Stomach. The gastric mucosa exhibited a folded, pink, and glossy appearance with intact epithelial linings. The non-glandular stomach portion was covered by stratified squamous epithelium, while the glandular part displayed mucus-secreting cylindrical cells without epithelial defects.

Small and Large Intestine. The small intestine’s mucosa was pale pink and shiny, forming well-defined villi covered with a single-layer prismatic epithelium. Goblet cells were abundant, with no epithelial abnormalities. Similarly, the large intestine’s mucosa was smooth, shiny, and grayish, lined with columnar epithelium rich in goblet exocrine cells located mainly in the crypts. Muscularis mucosae were intact in both segments.

Pancreas. Pancreatic lobular structures were preserved, with Langerhans’s islets and exocrine cells exhibiting normal nuclear morphology and cytoplasmic staining. The epithelial cells of the exocrine part of the gland contained basophilic nuclei located in the middle part, which was clear, with a sufficient amount of chromatin. Stroma was moderately vascularized, with no signs of pathology.

Liver. The liver was smooth and dark red, with a clear lobular structure and distinct hepatocyte boundaries. The shape and size were unchanged. Sinusoids were moderately plethoric, with hepatocyte nuclei displaying visible nucleoli and chromatin organization.

Spleen. The spleen retained a typical structure with clearly defined red and white pulp, and venous sinuses were moderately filled with erythrocytes. The surface of the organ was smooth, and the capsule was thin. Stroma contained thin trabeculae extending into the organ.

Kidneys and Urinary Bladder. The kidneys maintained a normal size, shape, and internal structure. The surface was brownish and smooth, and the capsule was thin and transparent. The glomeruli and tubules were intact, with no signs of pathology or inflammation. The bladder was filled with transparent urine, and its walls were structurally intact, with the clear delineation of mucosal, muscular, and adventitial layers.

In summary, macroscopic and histological analyses revealed no toxic effects following the single or repeated intravenous administration of WJ-MSC-SE33 across all doses studied. In addition, no effects on the reproductive system were detected. The ovaries (female) were oval, dense, grape-shaped, gray-pink in color, and compact. The testes (male) were of normal size and shape, moderately dense, and whitish.

#### 3.1.10. Local Tolerance at the Intravenous Injection Site of WJ-MSC-SE33 in Animals

Intravenous administrations of both genetically modified WJ-MSC-SE33 cells and native WJ-MSCs, alongside 0.9% of the NaCl solution for animals from the control group, were conducted via the tail vein. Observations of the tail skin and vein at the sites of both single and repeated administrations revealed no irritation, edema, infiltrates, or compactions. In the control and experimental groups, the intravenous administration of WJ-MSC-SE33 at doses of 0.5 × 10^7^, 1.25 × 10^7^, and 2.5 × 10^7^ cells/kg, regardless of administration frequency, did not result in the inflammatory infiltration of tissues surrounding the tail vein. The vein wall maintained uniform thickness with an intact endothelium (Figure 7). Thus, no local irritant effects were detected following either the single or repeated intravenous administration of WJ-MSC-SE33 at all doses studied in mice. This absence of adverse local reactions was comparable to that observed in control animals receiving 0.9% NaCl injections.

### 3.2. Sensitization Study of WJ-MSC-SE33 in Animals

No anaphylactic reactions were observed in animals administered WJ-MSC-SE33 at doses of 0.5 × 10^7^ and 2.5 × 10^7^ cells/kg compared to the controls sensitized with 0.6% HEW solution (Table 9). The absence of an allergenic response in both endolaryngeal and intravenous administration modes provides evidence that WJ-MSC-SE33 does not induce sensitization, even at doses five times higher than therapeutic levels.

### 3.3. Immunotoxicity Assessment of WJ-MSC-SE33 in Animals

#### 3.3.1. Humoral Immune Response in Mice: Hemagglutination Reaction

Following the repeated administration of WJ-MSC-SE33 and subsequent immunization with sheep erythrocytes, antibody titers in mouse serum showed no significant differences between animals receiving WJ-MSC-SE33 at therapeutic doses (0.5 × 10^7^ cells/kg) and control animals receiving 0.9% NaCl or native WJ-MSC injections (Figure 8A). However, a significant increase in antibody titers was observed in animals treated with WJ-MSC-SE33 at the 2.5 × 10^7^ cells/kg dose, indicating a dose-dependent humoral immune stimulation by genetically modified WJ-MSCs.

#### 3.3.2. Cellular Immune Response in Mice: DTH Reaction to Hapten

The DTH reaction was evaluated by measuring limb swelling after the injection of hapten in mice. Animals receiving multiple injections of WJ-MSC-SE33 at 0.5 × 10^7^ cells/kg exhibited increased limb swelling upon the administration of hapten compared to control animals receiving injections of 0.9% NaCl solution and native WJ-MSCs (Figure 8B). These findings suggest that WJ-MSC-SE33 administered at therapeutic doses can stimulate cellular immune responses, whereas higher doses (2.5 × 10^7^ cells/kg) did not induce such effects.

#### 3.3.3. Phagocytic Activity of Peritoneal Macrophages

The repeated intravenous administration of WJ-MSC-SE33 at 0.5 × 10^7^ cells/kg resulted in a significant reduction in the percentage of phagocytic cells compared to the control animals (0.9% NaCl). At the same time, a similar decrease in the number of phagocytic cells was observed in the control group that received injections of native WJ-MSCs (Figure 8C). In contrast, the administration of WJ-MSC-SE33 at 2.5 × 10^7^ cells/kg restored macrophage activity to control levels, suggesting potential immunosuppressive effects at therapeutic doses.

#### 3.3.4. Lymphocyte Blast Transformation

The analysis of spleen lymphocyte (splenocyte) proliferation after the course of repeated intravenous WJ-MSC-SE33 administration indicated that doses of 0.5 × 10^7^ and 2.5 × 10^7^ cells/kg significantly increased the number of blast lymphocytes during both spontaneous and LPS-induced proliferation in vitro (Figure 8D). Similar effects were observed in animals that received native WJ-MSCs at a dose of 2.5 × 10^7^ cells/kg. The data obtained indicate the stimulatory effect of WJ-MSC-SE33 on splenocyte activity, which may be due to the modulating effect on the cellular component of the immune response.

### 3.4. Immunogenicity of WJ-MSC-SE33 in Animals

To study the immunogenicity of WJ-MSC-SE33, levels of immunoglobulin M (IgM) and G (IgG) in mouse serum were assessed at regular intervals (days 14, 21, and 28) post-administration. The analysis revealed no significant changes in IgM levels in mice receiving the endolaryngeal administration of WJ-MSC-SE33 at doses of 0.5 × 10^7^ and 2.5 × 10^7^ cells/kg compared to both control groups. Similarly, no significant differences in IgG levels were detected between the groups receiving repeated intravenous injections of WJ-MSC-SE33 and control animals throughout the observation period (Figure 8E). These findings suggest that the repeated administration of WJ-MSC-SE33 by endolaryngeal and intravenous routes does not induce a dose-dependent immunogenic response in C57BL/6 mice, indicating minimal immunogenic potential for WJ-MSC-SE33 at the tested doses.

### 3.5. Study of the Antimicrobial Effect of WJ-MSC-SE33 Against S. aureus-Induced Bacterial Pneumonia in Mice

The antimicrobial activity of WJ-MSC-SE33 expressing the SE-33 peptide was evaluated in a model of bacterial pneumonia induced by intranasal infection with a suspension of *S. aureus*, a Gram-positive bacterium, which was selected as the causative agent of pneumonia due to its clinical significance as a leading cause of severe pulmonary infections, including acute lung injury [60]. This pathogen has been extensively studied in the context of MSC-based therapies for respiratory infections [61,62,63].

Within the first four days post-infection, mortality was observed in the control group receiving saline injections. In the control group injected with native WJ-MSCs, a slight but statistically non-significant reduction in mortality was observed compared to the saline-treated group (χ^2^ = 2.161, *p* = 0.142). Similarly, the group treated with WJ-MSC-SE33 at a dose of 0.5 × 10^7^ cells/kg exhibited mortality rates comparable to the group receiving native WJ-MSCs (χ^2^ = 2.155, *p* = 0.142). However, the administration of WJ-MSC-SE33 at doses of 1.0 × 10^7^ and 1.5 × 10^7^ cells/kg resulted in a significant reduction in mortality and improved survival compared to the saline-treated control group (χ^2^ = 3.784 and 3.871, respectively, *p* < 0.05; Figure 9A). Notably, no animal deaths were recorded in any experimental group beyond day 5.

The bacterial load, assessed as CFU in BALF, demonstrated the antibacterial effects of both native WJ-MSCs and WJ-MSC-SE33 on day 5 post-administration. CFU levels of *S. aureus* in the lavage fluid of mice treated with WJ-MSC-SE33 were approximately 12-fold lower compared to the saline-treated control group and 1.7-fold lower compared to the group receiving native WJ-MSCs. On day 8, CFU levels in the lavage fluid of mice treated with WJ-MSC-SE33 were approximately two-fold lower than those in mice treated with native WJ-MSCs (Figure 9B).

Analysis of leukocyte counts in BALF revealed a statistically significant reduction in leukocyte numbers in the groups treated with WJ-MSC-SE33 across all studied doses. By contrast, leukocyte counts in the lavage fluid of the two control groups showed no significant differences, suggesting that native WJ-MSCs did not significantly mitigate the inflammatory response associated with bacterial pneumonia. On day 5, leukocyte counts in the WJ-MSC-SE33-treated groups decreased by 18.98%, 20.34%, and 19.66% at doses of 0.5 × 10^7^, 1 × 10^7^, and 1.5 × 10^7^ cells/kg, respectively, compared to the control animals. On day 8, leukocyte counts decreased further by 21.79%, 22.84%, and 21.45% for the same doses of WJ-MSC-SE33, respectively, compared to both control groups (Figure 9C). These findings suggest that a reduction in pulmonary inflammatory edema is associated with *S. aureus* infection.

Histological and morphometric analyses of lung tissue following the repeated intravenous administration of WJ-MSC-SE33 were conducted in the context of bacterial pneumonia. A statistically significant reduction in lung mass coefficients was observed in mice treated with both native WJ-MSCs and WJ-MSC-SE33, with an average decrease of 10% compared to the saline-treated control group on both days 5 and 8 (Figure 9D). On day 5 of bacterial pneumonia development, the lungs of mice in the control group that received saline injections exhibited marked structural disruption. This disruption was characterized by interstitial edema localized in the peribronchial region and around the associated vessels, as well as diffuse inflammatory infiltrates in these areas. Additionally, the thickening of the alveolar septa was observed, along with areas of disrupted histoarchitecture in the respiratory region, alveolar filling with inflammatory cells, and hemorrhages (Figure 9E). By day 8, these pathological changes in lung tissue structure had intensified. In contrast, the lungs of control animals receiving injections of native WJ-MSCs displayed a reduction in the severity of peribronchial and perivascular edema, along with occasional inflammatory cells and small areas of hemorrhage in the alveolar lumen (Figure 9E).

Notably, no significant differences were observed among animals in the experimental groups that received injections of WJ-MSC-SE33 expressing the antimicrobial peptide SE-33 at all studied doses. Regardless of the dose or the duration of the period after the final administration, injections of WJ-MSC-SE33 consistently led to a reduction in the severity of both peribronchial and perivascular edema, as well as a decrease in inflammatory infiltrates. However, the complete resolution of these inflammatory signs was not achieved. The lung parenchyma of mice treated with WJ-MSC-SE33 presented a mosaic pattern. Regions with well-expanded alveoli with thin walls coexisted with areas of overstretched alveoli, which are frequently associated with alveolar wall thickening due to edema, capillary congestion, and residual inflammatory cell infiltration. Inflammatory cells were rarely observed within the alveolar lumina, and minor hemorrhages were also noted. Quantitative analysis revealed statistically significant reductions in inflammatory cell infiltration in pulmonary vessel walls, interstitial congestion, and hemorrhages in the lungs of mice treated with WJ-MSC-SE33 compared to saline-treated controls, indicating the therapeutic efficacy of this intervention. The results of the semi-quantitative histological analysis are presented in Table 10.

## 4. Discussion

In this study, we examined the general toxicity and immunotoxicity of genetically modified WJ-MSCs expressing the antimicrobial peptide SE-33. Additionally, we investigated the effect of genetically modified WJ-MSCs expressing the antimicrobial peptide SE-33 in a mouse model of bacterial pneumonia induced by *S. aureus*.

Our findings indicate that the administration of WJ-MSC-SE33 at different dosages elicits no adverse clinical manifestations of toxicity or deviations in body weight in experimental animals across the observation period. Moreover, no mortality was observed in groups receiving either single or repeated injections of genetically modified WJ-MSCs. Consistent with the data of other studies [64,65,66,67,68,69], animals that received injections of native WJ-MSCs showed no signs of acute toxicity or mortality. These results collectively suggest that WJ-MSC-SE33 does not induce acute toxicity or lethality.

The macroscopic and histological analyses of internal organs and tissues in animals receiving both therapeutic and supratherapeutic doses of WJ-MSC-SE33, as well as native WJ-MSCs, revealed no pathological alterations. Organ mass coefficients remained unchanged following both single and repeated administrations across all dosages. Furthermore, neither WJ-MSC-SE33 nor native WJ-MSCs exerted any observable neurotoxic effects or impaired CNS function in the animals. These results demonstrate that WJ-MSC-SE33 does not cause organ damage, hypertrophy, atrophy, or other unfavorable morphological changes, supporting the findings from prior studies with native WJ-MSCs [64,68,69].

The analysis of hematological parameters, including hemoglobin, erythrocyte, leukocyte, and platelet counts, revealed no significant changes across groups receiving WJ-MSC-SE33 at different doses or native WJ-MSCs. This suggests that neither the expression of the SE-33 peptide by genetically modified cells nor WJ-MSC administration affects hematopoiesis or cellular integrity, as indicated by the unaltered blood cell profiles. Additionally, serum biochemistry analysis, including liver and kidney function markers (e.g., total protein, albumin, urea, creatinine, and cholesterol), showed no statistically significant alterations post-administration of WJ-MSC-SE33. All serum parameters remained within the normal ranges observed in animals administered native WJ-MSCs, which is consistent with other studies [67,68]. The absence of significant changes in renal and hepatic function parameters suggests that WJ-MSC-SE33 does not induce hepatotoxicity or nephrotoxicity. Furthermore, studies have shown that the intravenous administration of MSCs may contribute to kidney repair and mitigate glomerular abnormalities and liver fibrosis [70,71,72,73]. Notably, no signs of local intolerance, such as inflammation or tissue damage at the injection site, were observed following either the single or repeated intravenous administration of WJ-MSC-SE33.

The study of immunotoxicity and immunogenicity in experimental animals indicated that WJ-MSC-SE33, at both a therapeutic dose (0.5 × 10^7^ cells/kg) and a five-fold excess dose (2.5 × 10^7^ cells/kg), did not induce anaphylactic reactions, demonstrating a lack of sensitizing potential. The high-dose administration of WJ-MSC-SE33 to mice significantly stimulated humoral immunity, evidenced by elevated serum antibody titers. In contrast, therapeutic doses (0.5 × 10^7^ cells/kg) and native WJ-MSCs at a dose of 2.5 × 10^7^ cells/kg showed no such effect. This phenomenon may be associated with elevated SE-33 peptide levels following higher cell dosage administration, such as SE-33, a retroanalog of human cathelicidin LL-37, which may provoke immune activation at increased concentrations [74]. Furthermore, the repeated administration of WJ-MSC-SE33 did not significantly alter IgM or IgG serum levels relative to control animals or animals injected with native WJ-MSCs, indicating that genetically modified WJ-MSCs possess no significant immunogenic potential under the studied conditions [75,76,77].

We also found that WJ-MSC-SE33 at a therapeutic dose (0.5 × 10^7^ cells/kg) slightly but significantly stimulated cellular immunity, as indicated by an increased DTH response to antigen. However, this effect was not observed with a five-fold excess dose (2.5 × 10^7^ cells/kg) for both WJ-MSC-SE33 and native WJ-MSCs. Notably, the repeated administration of WJ-MSC-SE33 at a therapeutic dose and native WJ-MSCs led to a modest decrease in phagocytic peritoneal macrophage counts, while the phagocytic activity of macrophages in animals receiving the excess dose of WJ-MSC-SE33 was restored to levels observed in control animals. It has been established that interactions with MSCs and cathelicidin LL-37 can enhance macrophage phagocytic and bactericidal activity [78,79,80]. While our results may initially appear contradictory, evidence suggests that MSCs can both attenuate macrophage phagocytosis and induce apoptosis in these cells [81,82,83]. Finally, we observed a slight but statistically significant increase in splenic lymphocyte proliferation following the administration of both WJ-MSC-SE33 and native WJ-MSCs at therapeutic and supratherapeutic doses. Although MSCs are generally considered to inhibit lymphocyte proliferation in vitro, studies have shown that co-culturing non-activated allogeneic MSCs with splenocytes can stimulate splenocyte proliferation, aligning with our findings [83,84,85]. Taken together, these results support the established immunomodulatory effects of MSCs on immune system components [72]. Thus, our study demonstrates that genetically modified WJ-MSCs expressing SE-33 show the absence of significant toxicity or immunogenicity. Although xenogeneic experimental models involving the transplantation of human cells into immunocompetent mice have certain limitations, including the potential for cross-species incompatibilities, this model was selected in our study because preclinical studies utilizing the human cell product intended for clinical use are known to provide valuable insights and are often essential for the regulatory approval of clinical trial protocols [86]. Human MSCs (including those derived from birth-associated tissues) are known for their immunomodulatory properties, which allow them to interact with both innate and adaptive immune cells [87,88]. These properties not only contribute to their therapeutic efficacy but also minimize the risk of eliciting an allogeneic immune response when administered in vivo [87,89]. This characteristic supports the safe use of human MSCs in immunocompetent mouse models without triggering significant inflammatory responses, as reported in previous studies, which showed no immune-mediated side effects such as fever, joint swelling, skin rash, or increased systemic inflammation in fully immunocompetent C57BL/6 mice following human MSC injection [9,70,88,90]. Thus, our findings are consistent with these studies, demonstrating that human MSCs exhibit low immunogenicity and do not provoke significant inflammatory responses in immunocompetent mice, as observed in the present study [70,90].

The rising prevalence of antibiotic-resistant *S. aureus* strains, particularly methicillin-resistant *S. aureus* (MRSA), underscores the urgent need for alternative therapeutic strategies, such as cell-based approaches [91]. Unlike *P. aeruginosa* or *K. pneumonia*, Gram-negative pathogens commonly associated with pneumonia, *S. aureus*-induced infections, are characterized by excessive inflammatory responses, extensive lung tissue damage, and a high risk of complications, including sepsis and ARDS [60]. Previous studies have demonstrated the therapeutic potential of MSC-based interventions in *S. aureus*-induced pneumonia, particularly in reducing bacterial load and mitigating excessive inflammation [61,62,63]. However, these studies have focused on adipose tissue-derived MSCs, while data on the efficacy of WJ-MSCs in this context remain limited. Given these factors, an in vivo model of bacterial lung infection induced by *S. aureus* in C57BL/6 mice was chosen to evaluate the antimicrobial effect of WJ-MSC-SE33 expressing the SE-33 peptide. MSCs have been demonstrated to predominantly accumulate in the lungs after intravenous administration due to the pulmonary first-pass effect, making them important for the cell therapy of diseased or injured lungs [65,66,92,93]. Our findings demonstrate that the repeated administration of WJ-MSC-SE33 to mice with bacterial pneumonia exerts a potent antibacterial effect, accompanied by a significant reduction in mortality rates. Specifically, 8 days after the initiation of treatment with WJ-MSC-SE33, there was a 24-fold and 2.5-fold decrease in the CFU of *S. aureus* in the BALF of experimental animals compared to the groups receiving saline injections and native WJ-MSCs, respectively. However, the complete clearance of *S. aureus* from the lavage of the experimental animals was not achieved. The observed suppression of bacterial growth in the BALF of animals treated with native WJ-MSCs may be attributed to the paracrine secretion of AMPs, particularly LL-37, and active extracellular microvesicles released by the WJ-MSCs [20,29,61]. LL-37 is known to directly bind to lipopolysaccharides and inactivate them, thereby facilitating the disruption of the outer bacterial membrane. For instance, LL-37 secreted by MSCs has been shown to enhance the efficacy of antibiotics, promote pathogen killing, and inhibit bacterial growth in pulmonary infection models of cystic fibrosis caused by *P. aeruginosa*, *S. aureus*, and *S. pneumoniae* [61]. Our in vitro findings indicate that SE-33 predominantly accumulates within WJ-MSC-SE33 cells, with minimal extracellular secretion under normal conditions. Quantitative analysis also revealed that the concentration of SE-33 in supernatants (0.75 ng/mL) was lower than the levels of LL-37 (2–3 ng/mL) secreted by native MSCs reported in other studies [20,29]. These findings suggest that the antimicrobial activity observed in our in vivo study may be attributed to localized cellular release mechanisms rather than the continuous secretion of the SE-33 peptide. We propose that SE-33 secretion could be mediated by cell-to-cell contract or triggered by pathogen-associated molecules, such as lipopolysaccharides, which are known to enhance LL-37 secretion in native MSCs [20,29]. Additionally, SE-33 peptide release may occur due to the death and lysis of WJ-MSC-SE33. The lysis of these cells would likely facilitate the release of intracellular SE-33 directly at the site of infection, thereby contributing to the clearance of *S. aureus* and reducing the bacterial load in the lungs of mice with induced pulmonary pneumonia. However, little is known about the behavior of MSCs (proliferation, differentiation, or cell death) and fate following their administration in vivo [94]. Further studies are required to investigate the regulatory pathways involved in SE-33 peptide secretion and to validate the proposed release mechanisms in vivo.

In addition to its antimicrobial effects, the therapeutic intervention with WJ-MSC-SE33 alleviated the severity of lung inflammation during the early stages of bacterial pneumonia, as demonstrated by histological and morphometric analyses. It also led to an average 22% reduction in leukocyte counts in the lungs. Histological examinations revealed reductions in interstitial edema and inflammatory infiltrates surrounding the bronchi, along with small areas of thickened alveolar walls. MSCs are known to mitigate the progression of infectious disease by inhibiting the epithelial–mesenchymal transition, promoting lung fluid clearance, and stimulating the phagocytic activity of alveolar macrophages [29,95,96]. The observed reduction in alveolar inflammation in this study is likely the result of the immunomodulatory effects of MSCs on both innate and adaptive immune cells, which is further enhanced by the antimicrobial effects of the SE-33 peptide [97,98,99]. Recent studies align with our findings, demonstrating that the intravenous administration of LL-37 can improve the survival of septic mice through mechanisms such as the suppression of pro-inflammatory macrophage pyroptosis and the release of neutrophil extracellular traps and antimicrobial microvesicles (ectosomes) from neutrophils [100,101]. Additionally, LL-37 has been shown to alleviate sepsis-induced acute lung injury by downregulating the expression of key components of the pyroptotic pathway, including NLRP3, caspase-1, and GSDMD, as well as downstream inflammatory factors in alveolar epithelial cells [102]. Our findings are also consistent with a recent study showing that the administration of engineered MSCs expressing a fusion protein composed of a 21 kDa bioactive fragment of the bactericidal/permeability-increasing protein (BPI21) and LL-37 significantly reduced the inflammatory response, mitigated organ damage, and improved survival in mice with *S. aureus*-induced sepsis [43]. Thus, our results confirm the high efficacy and advantage of treating infectious pneumonia using genetically modified cells expressing AMPs over therapy with native MSCs.

## 5. Conclusions

In this study, we demonstrated that genetically modified WJ-MSCs expressing the antimicrobial peptide SE-33 do not induce acute or chronic toxicity, dysfunction, or histopathological alterations in major organs following intravenous administration. Importantly, no significant signs of immunotoxicity or immunogenicity were detected when compared to native WJ-MSCs. Additionally, no behavioral abnormalities or signs of stress were noted in the experimental animals throughout the observation period, indicating that the genetically modified WJ-MSCs were well tolerated, similar to native cells. Furthermore, we demonstrated that WJ-MSCs expressing the SE-33 peptide exhibited significant antimicrobial activity against *S. aureus*-induced bacterial lung infection. Genetically modified WJ-MSCs significantly decreased animal mortality associated with bacterial pneumonia and improved the clearance of bacterial pathogens from the lungs, demonstrating the effectiveness of these cells compared to native WJ-MSCs. Additionally, reductions in leukocyte counts and inflammatory infiltrates in the lung tissue suggest that SE-33 possesses immunomodulatory properties that enhance the natural antimicrobial effects of MSCs.

Taken together, these findings support the safety profile of WJ-MSC-SE33 as a promising candidate for the development of cell-based therapies. Future studies should focus on evaluating the biodistribution and accumulation of WJ-MSCs expressing the SE-33 peptide. Additionally, comprehensive studies are necessary to investigate the tumorigenic and oncogenic potential of WJ-MSC-SE33 cells.

## Figures and Tables

**Figure 1 cells-14-00341-f001:**
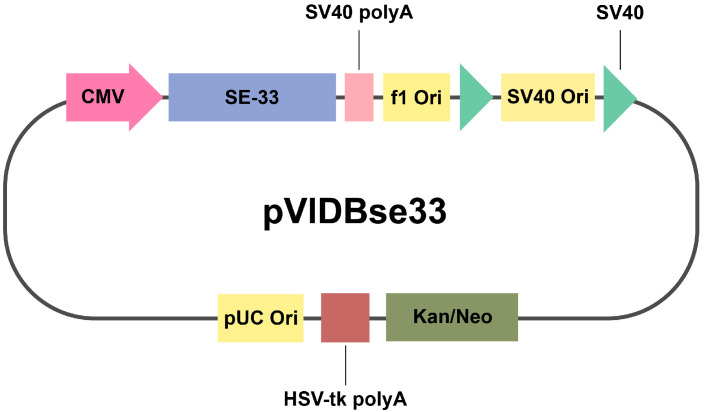
Plasmid map of the constructed vector.

**Figure 2 cells-14-00341-f002:**
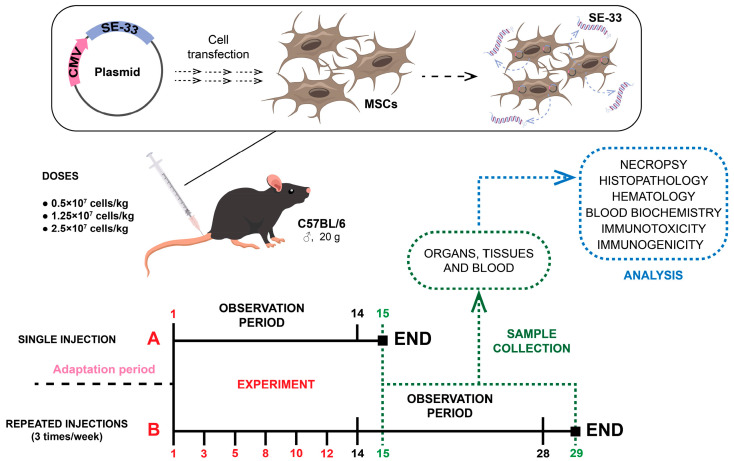
General scheme for toxicity evaluation of genetically modified WJ-MSCs after single (**A**) and repeated (**B**) administrations.

**Figure 3 cells-14-00341-f003:**
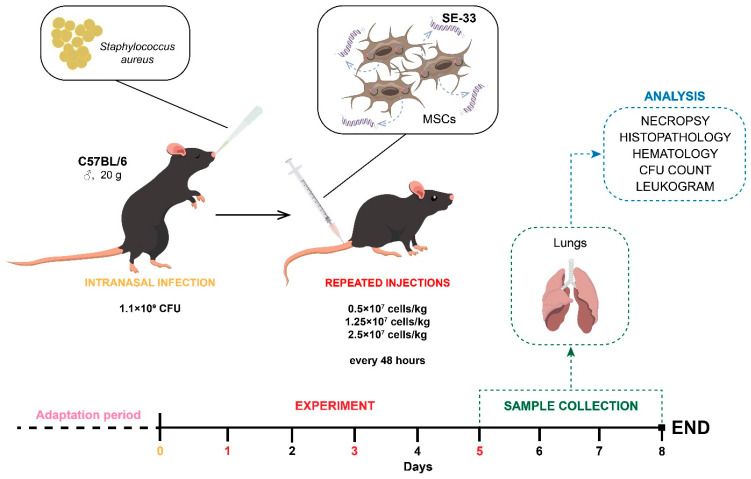
Experimental design to evaluate the ability of WJ-MSC-SE33 to exhibit the antimicrobial effect in a murine model of bacterial lung infection.

**Figure 4 cells-14-00341-f004:**
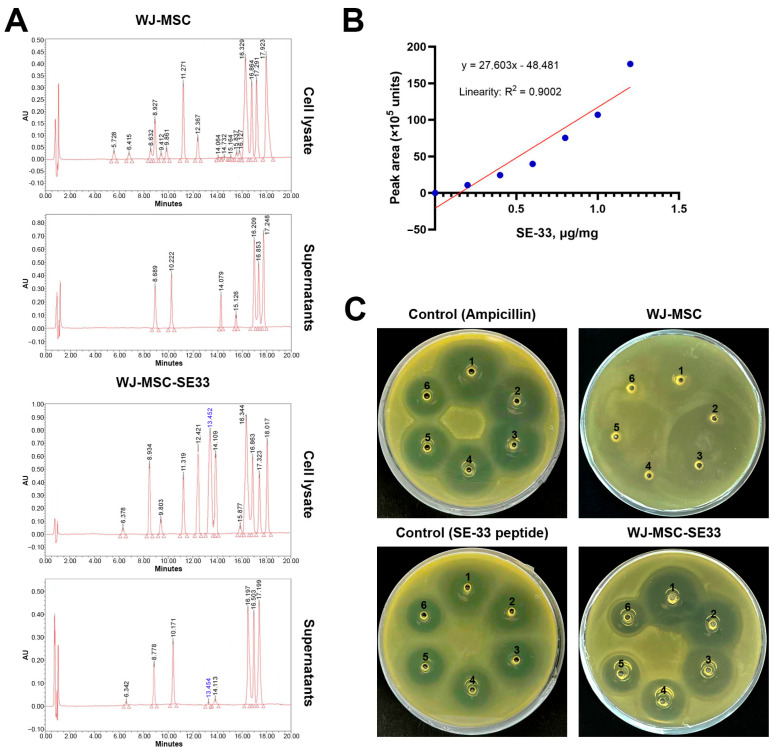
Expression and functional analysis of the SE-33 peptide using genetically modified WJ-MSCs. (**A**) Representative chromatograms showing the SE-33 peptide expression in WJ-MSCs and WJ-MSC-SE33. The peak concentrations corresponding to the SE-33 peptide are indicated in blue. (**B**) A calibration curve illustrating the correlation between the peak area and the concentration of the SE-33 peptide constructed using SE-33 peptide solutions. (**C**) The analysis of antimicrobial activity against *E. coli* for native and genetically modified WJ-MSCs expressing the SE-33 peptide. Ampicillin (25 μg/mL), serial dilutions: 1—stock solution (40 μL); 2—1:2; 3—1:4; 4—1:8; 5—1:16; 6—1:32. SE-33 peptide (1 mg/mL), serial dilutions: 1—stock solution (40 μL); 2—1:2; 3—1:4; 4—1:8; 5—1:16; 6—1:32. WJ-MSCs lysate: 1—4 × 10^5^ cells; 2—2 × 10^5^ cells; 3—1 × 10^5^ cells; 4—5 × 10^4^ cells; 5—2.5 × 10^4^ cells; 6—1.25 × 10^4^ cells. WJ-MSC-SE33 cell lysate: 1—4 × 10^5^ cells; 2—2 × 10^5^ cells; 3—1 × 10^5^ cells; 4—5 × 10^4^ cells; 5—2.5 × 10^4^ cells; 6—1.25 × 10^4^ cells.

**Figure 5 cells-14-00341-f005:**
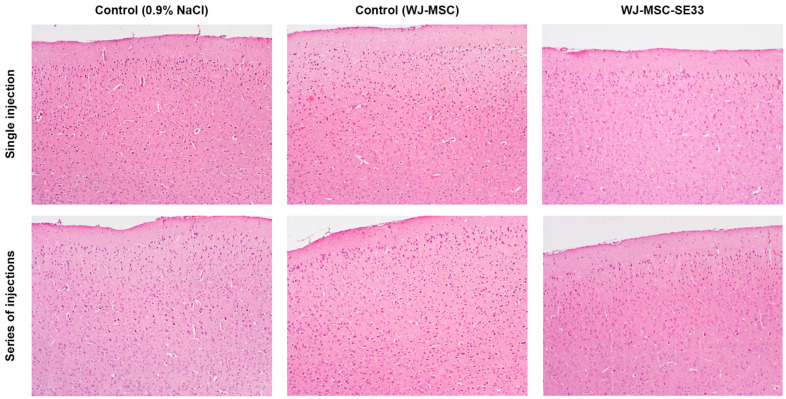
Mouse brain sections with cortical layers post single and repeat injections of WJ-MSC-SE33. Control groups: 0.9% NaCl and native WJ-MSCs at 2.5 × 10^7^ cells/kg. Experimental group: WJ-MSC-SE33 at 2.5 × 10^7^ cells/kg. Magnification ×100. Hematoxylin and eosin staining.

**Figure 6 cells-14-00341-f006:**
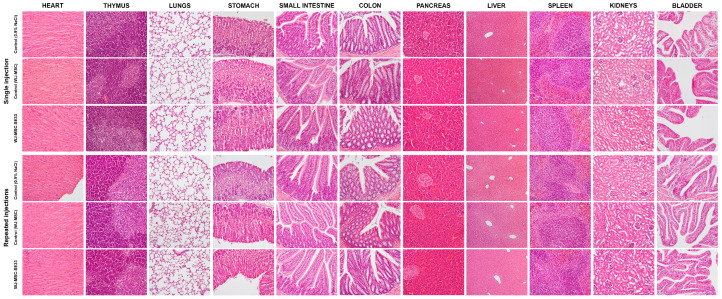
Mouse organ sections post single and repeat injections of WJ-MSC-SE33: the heart (myocardium), thymus (structural integrity), lungs (alveolar architecture), stomach wall, small intestine (mucosal architecture), large intestine/colon (mucosal architecture), pancreas, liver, spleen, kidneys, and urinary bladder. Control groups: 0.9% NaCl and native WJ-MSCs at 2.5 × 10^7^ cells/kg. Experimental group: WJ-MSC-SE33 at 2.5 × 10^7^ cells/kg. Magnification ×200. Hematoxylin and eosin staining.

**Figure 7 cells-14-00341-f007:**
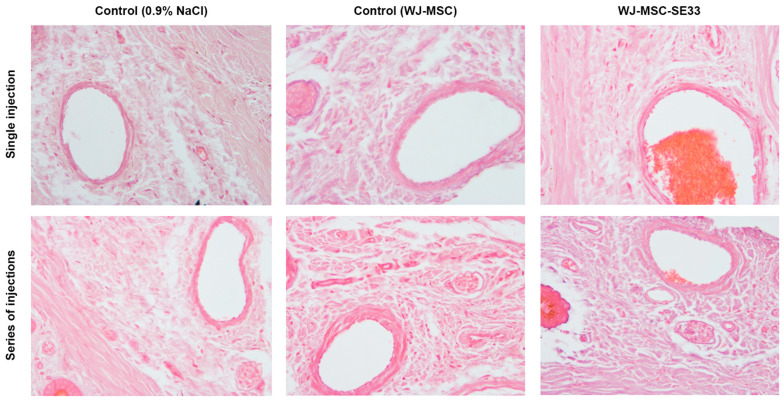
Histological sections of the mouse tail vein after single and repeated injections of WJ-MSC-SE33: Control groups: 0.9% NaCl and native WJ-MSCs at 2.5 × 10^7^ cells/kg. Experimental groups: WJ-MSC-SE33 at 2.5 × 10^7^ cells/kg. Magnification ×400; Hematoxylin and eosin staining.

**Figure 8 cells-14-00341-f008:**
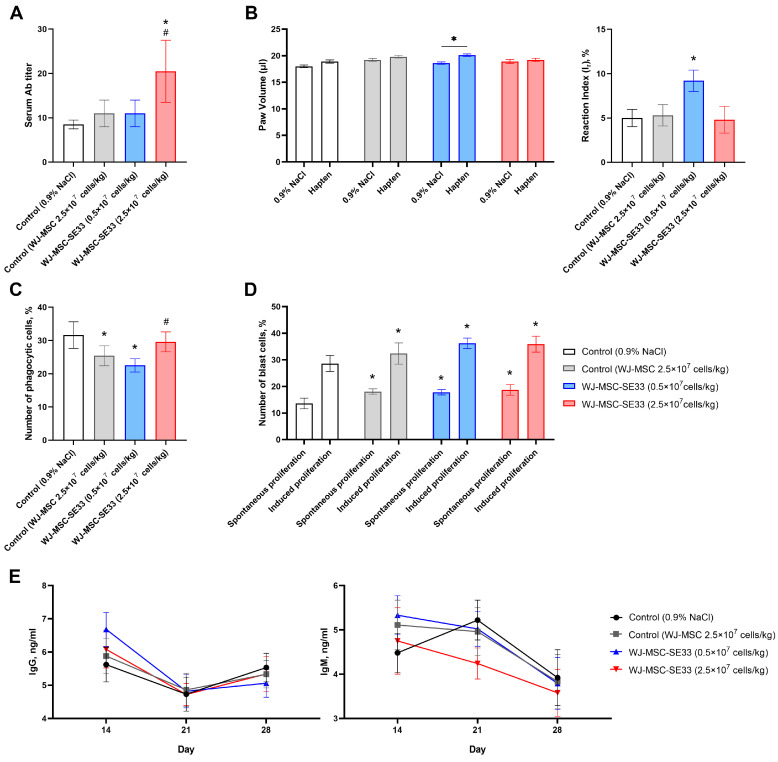
Evaluation of immunotoxicity and immunogenic effects following the repeated administration of WJ-MSC-SE33 in animals. (**A**) Assessment of the humoral immune response through hemagglutination in mice receiving WJ-MSC-SE33. (**B**) Evaluation of anti-inflammatory activity following WJ-MSC-SE33 administration. (**C**) Analysis of phagocytic activity in peritoneal macrophages post-injection with WJ-MSC-SE33. (**D**) Functional assessment of spleen lymphocyte activity through the blast transformation reaction post-WJ-MSC-SE33 administration. (**E**) The study of WJ-MSC-SE33 immunogenicity, quantifying serum IgG and IgM in treated animals. Kruskal–Wallis test, mean ± SEM. * *p* < 0.05 relative to the control group (0.9% NaCl), # *p* < 0.05 relative to the control group (WJ-MSCs at 2.5 × 10^7^ cells/kg).

**Figure 9 cells-14-00341-f009:**
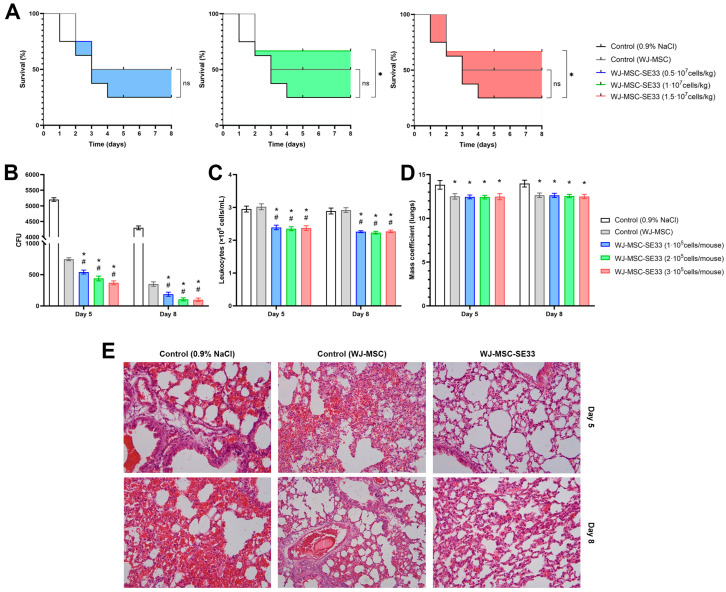
The antimicrobial properties of WJ-MSC-SE33 expressing the SE-33 peptide in mice with *S. aureus*-induced pneumonia. (**A**) Survival curves (Kaplan–Meier) of mice on day 5 post-infection. Log Rank (Mantel–Cox), * *p* < 0.05 compared to the control group (0.9% NaCl), ns—not significant. (**B**) Bacterial load in BALF after WJ-MSC-SE33 administration. (**C**) Leukocyte counts in BALF after WJ-MSC-SE33 administration. (**D**) Lung mass coefficients following WJ-MSC-SE33 administration. Kruskal–Wallis test, M ± SEM. * *p* < 0.05 compared to the control group (0.9% NaCl), # *p* < 0.05 compared to the control group (native WJ-MSC). (**E**) Lung sections on days 5 and 8. Hematoxylin and eosin staining. Magnification ×200.

**Table 1 cells-14-00341-t001:** Body weight (g) of mice following a single intravenous administration of genetically modified MSCs expressing SE-33 (mean ± SEM).

Observation Period	Sex	Control (0.9% NaCl) (n = 12)	Control (WJ-MSC 2.5 × 10^7^ Cells/kg) (n = 12)	WJ-MSC-SE33		
				0.5 × 10^7^ Cells/kg (n = 12)	1.25 × 10^7^ Cells/kg (n = 12)	2.5 × 10^7^ Cells/kg (n = 12)
0 days	Male	18.92 ± 0.21	19.00 ± 0.21	18.92 ± 0.23	18.92 ± 0.28	18.88 ± 0.18
	Female	18.88 ± 0.25	18.54 ± 0.22	18.83 ± 0.26	18.54 ± 0.18	18.79 ± 0.23
7 days	Male	19.17 ± 0.23	19.21 ± 0.23	19.25 ± 0.23	19.17 ± 0.25	19.21 ± 0.14
	Female	19.04 ± 0.28	18.75 ± 0.21	19.13 ± 0.24	18.75 ± 0.16	19.13 ± 0.23
14 days	Male	19.38 ± 0.22	19.42 ± 0.23	19.50 ± 0.17	19.50 ± 0.20	19.50 ± 0.14
	Female	19.33 ± 0.30	19.13 ± 0.16	19.17 ± 0.20	19.04 ± 0.18	19.54 ± 0.20

**Table 2 cells-14-00341-t002:** Rectal temperature (°C) of mice following a single intravenous administration of genetically modified MSCs expressing SE-33 (mean ± SEM).

Observation Period	Sex	Control (0.9% NaCl) (n = 12)	Control (WJ-MSC 2.5 × 10^7^ Cells/kg) (n = 12)	WJ-MSC-SE33		
				0.5 × 10^7^ Cells/kg (n = 12)	1.25 × 10^7^ Cells/kg (n = 12)	2.5 × 10^7^ Cells/kg (n = 12)
0 days	Male	37.58 ± 0.07	37.55 ± 0.05	37.56 ± 0.06	37.64 ± 0.05	37.63 ± 0.06
	Female	37.66 ± 0.07	37.64 ± 0.05	37.57 ± 0.06	37.61 ± 0.06	37.63 ± 0.05
7 days	Male	37.64 ± 0.07	37.64 ± 0.07	37.58 ± 0.07	37.62 ± 0.07	37.62 ± 0.06
	Female	37.53 ± 0.05	37.59 ± 0.08	37.58 ± 0.05	37.69 ± 0.07	37.59 ± 0.07
14 days	Male	37.54 ± 0.07	37.58 ± 0.06	37.63 ± 0.05	37.54 ± 0.07	37.58 ± 0.06
	Female	37.58 ± 0.06	37.53 ± 0.05	37.60 ± 0.07	37.61 ± 0.05	37.56 ± 0.06

**Table 3 cells-14-00341-t003:** Effect of a single intravenous administration of WJ-MSC-SE33 on the behavioral parameters of mice in the open field test (M ± SEM).

Observation Period	Sex	Control (0.9% NaCl) (n = 12)	Control (WJ-MSC 2.5 × 10^7^ Cells/kg) (n = 12)	WJ-MSC-SE33		
				0.5 × 10^7^ Cells/kg (n = 12)	1.25 × 10^7^ Cells/kg (n = 12)	2.5 × 10^7^ Cells/kg (n = 12)
Horizontal activity (duration, sec)						
0 days	Male	208.68 ± 5.21	208.65 ± 5.50	202.1 ± 3.62	210.49 ± 5.96	206.08 ± 5.02
	Female	198.22 ± 6.09	203.31 ± 4.15	203.26 ± 5.96	203.46 ± 5.82	201.24 ± 6.41
14 days	Male	205.13 ± 5.24	209.73 ± 4.58	208.99 ± 4.34	202.79 ± 6.00	210.49 ± 5.23
	Female	205.16 ± 5.84	202.33 ± 5.88	198.05 ± 5.68	205.06 ± 5.71	196.53 ± 6.07
Vertical activity (number of vertical stands)						
0 days	Male	29.00 ± 0.55	28.33 ± 0.57	28.42 ± 0.47	28.83 ± 0.42	28.00 ± 0.59
	Female	28.67 ± 0.59	28.08 ± 0.47	27.83 ± 0.47	28.50 ± 0.50	28.42 ± 0.48
14 days	Male	28.25 ± 0.48	29.17 ± 0.53	28.67 ± 0.50	28.17 ± 0.46	28.58 ± 0.43
	Female	28.17 ± 0.49	28.33 ± 0.47	28.92 ± 0.53	28.58 ± 0.53	28.50 ± 0.48
Grooming duration, sec						
0 days	Male	3.31 ± 0.06	3.29 ± 0.06	3.29 ± 0.06	3.23 ± 0.05	3.23 ± 0.04
	Female	3.25 ± 0.06	3.21 ± 0.06	3.30 ± 0.05	3.26 ± 0.05	3.27 ± 0.05
14 days	Male	3.29 ± 0.05	3.32 ± 0.08	3.24 ± 0.04	3.22 ± 0.07	3.30 ± 0.06
	Female	3.31 ± 0.06	3.27 ± 0.05	3.30 ± 0.06	3.32 ± 0.05	3.23 ± 0.04
Duration of freezing reaction, sec						
0 days	Male	0.67 ± 0.03	0.62 ± 0.03	0.65 ± 0.04	0.68 ± 0.04	0.62 ± 0.04
	Female	0.67 ± 0.05	0.72 ± 0.04	0.69 ± 0.03	0.69 ± 0.04	0.63 ± 0.05
14 days	Male	0.70 ± 0.03	0.65 ± 0.04	0.69 ± 0.04	0.63 ± 0.03	0.64 ± 0.04
	Female	0.68 ± 0.04	0.67 ± 0.04	0.63 ± 0.04	0.67 ± 0.04	0.69 ± 0.04
Duration of sniffing reaction, sec						
0 days	Male	44.69 ± 3.09	45.33 ± 2.11	43.46 ± 2.11	44.33 ± 3.04	48.13 ± 2.27
	Female	45.04 ± 2.75	43.96 ± 2.62	48.93 ± 2.98	45.34 ± 1.74	44.85 ± 2.31
14 days	Male	46.3 ± 2.76	46.87 ± 1.84	50.01 ± 2.49	47.09 ± 3.02	49.38 ± 2.58
	Female	43.46 ± 2.32	47.71 ± 2.88	44.59 ± 2.9	48.33 ± 2.19	44.26 ± 2.82

**Table 4 cells-14-00341-t004:** Effect of a single intravenous WJ-MSC-SE33 administration on the morphological composition of the blood in mice (M ± SEM).

Observation Period	Sex	Control (0.9% NaCl) (n = 6)	Control (WJ-MSC 2.5 × 10^7^ Cells/kg) (n = 6)	WJ-MSC-SE33		
				0.5 × 10^7^ Cells/kg (n = 6)	1.25 × 10^7^ Cells/kg (n = 6)	2.5 × 10^7^ Cells/kg (n = 6)
Hemoglobin, g/L						
15 days	Male	143.83 ± 3.93	131.17 ± 6.93	142.50 ± 1.73	144.67 ± 1.45	145.33 ± 3.82
	Female	142.33 ± 3.14	140.17 ± 3.38	143.83 ± 1.49	146.83 ± 2.12	144.00 ± 4.66
Erythrocytes, ×10^12^/L						
15 days	Male	8.43 ± 0.26	7.82 ± 0.42	8.50 ± 0.12	8.78 ± 0.11	8.68 ± 0.25
	Female	8.32 ± 0.23	8.33 ± 0.23	8.54 ± 0.15	8.71 ± 0.19	8.59 ± 0.27
Leukocytes, ×10^9^/L						
15 days	Male	7.25 ± 0.70	6.50 ± 0.96	6.68 ± 0.92	5.95 ± 0.92	6.08 ± 0.89
	Female	6.72 ± 0.95	5.10 ± 0.87	5.13 ± 0.63	5.37 ± 0.99	5.02 ± 0.76
Platelets, ×10^9^/L						
15 days	Male	655.67 ± 37.83	650.00 ± 23.56	597.33 ± 48.78	619.33 ± 43.46	653.00 ± 32.06
	Female	664.67 ± 50.31	772.17 ± 35.20	621.00 ± 70.47	702.00 ± 37.60	661.83 ± 40.47
ESR, mm/h						
15 days	Male	1.17 ± 0.17	1.17 ± 0.17	1.33 ± 0.33	1.17 ± 0.17	1.17 ± 0.17
	Female	1.50 ± 0.34	1.17 ± 0.17	1.17 ± 0.17	1.17 ± 0.17	1.33 ± 0.21

**Table 5 cells-14-00341-t005:** Effect of a single intravenous WJ-MSC-SE33 administration on leukocyte count in mice (M ± SEM).

Observation Period	Sex	Control (0.9% NaCl) (n = 6)	Control (WJ-MSC 2.5 × 10^7^ Cells/kg) (n = 6)	WJ-MSC-SE33		
				0.5 × 10^7^ Cells/kg (n = 6)	1.25 × 10^7^ Cells/kg (n = 6)	2.5 × 10^7^ Cells/kg (n = 6)
Band neutrophils, %						
15 days	Male	0	0	0.17±0.17	0.17±0.17	0
	Female	0.17 ± 0.17	0	0	0.17 ± 0.17	0
Segmented neutrophils, %						
15 days	Male	19.17 ± 1.14	20.00 ± 3.89	20.83 ± 2.70	22.67 ± 2.39	21.00 ± 2.13
	Female	18.67 ± 1.82	21.00 ± 3.59	21.33 ± 1.82	21.00 ± 2.29	19.00 ± 1.69
Eosinophils, %						
15 days	Male	2.67 ± 0.49	1.67 ± 0.71	2.50 ± 0.56	2.00 ± 0.58	2.83 ± 0.48
	Female	2.33 ± 0.67	2.17 ± 0.60	2.50 ± 0.85	2.00 ± 0.86	2.67 ± 0.49
Basophils, %						
15 days	Male	0	0	0	0	0
	Female	0	0	0	0	0
Monocytes, %						
15 days	Male	3.50 ± 0.62	3.33 ± 0.76	3.17 ± 0.95	2.83 ± 0.83	3.50 ± 0.67
	Female	3.17 ± 0.87	3.50 ± 0.72	2.50 ± 0.56	3.00 ± 0.97	2.83 ± 0.48
Lymphocytes, %						
15 days	Male	74.67 ± 1.45	75.00 ± 3.40	73.33 ± 2.74	72.33 ± 2.93	72.67 ± 2.39
	Female	75.67 ± 1.61	73.33 ± 4.38	73.67 ± 1.23	73.83 ± 2.15	75.50 ± 1.95

**Table 6 cells-14-00341-t006:** Effect of a single intravenous WJ-MSC-SE33 administration on the biochemical blood parameters in mice (M ± SEM).

Observation Period	Sex	Control (0.9% NaCl) (n = 6)	Control (WJ-MSC 2.5 × 10^7^ Cells/kg) (n = 6)	WJ-MSC-SE33		
				0.5 × 10^7^ Cells/kg (n = 6)	1.25 × 10^7^ Cells/kg (n = 6)	2.5 × 10^7^ Cells/kg (n = 6)
Total protein, g/L						
15 days	Male	56.55 ± 1.20	57.37 ± 1.42	56.63 ± 1.20	57.02 ± 1.24	56.37 ± 1.63
	Female	56.34 ± 1.52	58.68 ± 1.32	57.38 ± 1.49	58.54 ± 1.28	57.07 ± 1.26
Albumin, g/L						
15 days	Male	34.09 ± 0.98	33.98 ± 0.89	32.12 ± 0.97	33.40 ± 0.87	33.28 ± 1.09
	Female	34.44 ± 1.13	34.23 ± 0.17	33.65 ± 0.86	35.19 ± 1.69	33.80 ± 1.28
Urea, mmol/L						
15 days	Male	6.62 ± 0.10	6.57 ± 0.09	6.63 ± 0.08	6.63 ± 0.12	6.69 ± 0.10
	Female	6.69 ± 0.12	6.64 ± 0.11	6.56 ± 0.07	6.57 ± 0.10	6.68 ± 0.12
Creatinine, µmol/L						
15 days	Male	74.80 ± 0.94	74.36 ± 1.15	73.66 ± 1.01	72.91 ± 0.75	75.09 ± 0.77
	Female	74.89 ± 1.05	74.53 ± 0.88	75.30 ± 0.66	74.31 ± 0.76	73.08 ± 0.92
Total cholesterol, mmol/L						
15 days	Male	2.00 ± 0.12	2.17 ± 0.06	2.02 ± 0.08	2.06 ± 0.06	1.98 ± 0.13
	Female	1.95 ± 0.11	1.85 ± 0.10	2.00 ± 0.05	2.01 ± 0.12	2.07 ± 0.16
Triglycerides, mmol/L						
15 days	Male	1.02 ± 0.12	1.08 ± 0.14	1.05 ± 0.11	1.08 ± 0.12	1.06 ± 0.13
	Female	1.07 ± 0.16	1.06 ± 0.10	1.10 ± 0.11	1.05 ± 0.09	1.12 ± 0.012
Aspartate aminotransferase, U/L						
15 days	Male	225.30 ± 22.29	242.07 ± 26.58	261.48 ± 16.69	257.86 ± 25.55	248.88 ± 19.64
	Female	247.72 ± 28.14	266.51 ± 33.87	257.74 ± 42.20	277.79 ± 32.04	256.82 ± 26.73
Alanine aminotransferase, U/L						
15 days	Male	54.24 ± 6.33	57.36 ± 9.81	63.86 ± 5.81	66.87 ± 7.38	67.66 ± 8.04
	Female	62.85 ± 8.44	70.89 ± 10.76	62.67 ± 3.47	72.85 ± 8.71	69.62 ± 8.29
Alkaline phosphatase, U/L						
15 days	Male	105.93 ± 16.46	103.74 ± 5.92	106.45 ± 11.34	101.21 ± 6.22	101.93 ± 10.01
	Female	106.95 ± 9.39	106.03 ± 14.07	111.81 ± 6.46	113.64 ± 6.22	110.76 ± 7.49
Glucose, mmol/L						
15 days	Male	4.59 ± 0.10	4.57 ± 0.10	4.52 ± 0.09	4.62 ± 0.10	4.53 ± 0.06
	Female	4.61 ± 0.13	4.60 ± 0.06	4.54 ± 0.09	4.58 ± 0.06	4.56 ± 0.09
Total bilirubin, µmol/L						
15 days	Male	1.46 ± 0.36	1.98 ± 0.58	2.19 ± 0.58	1.69 ± 0.56	1.61 ± 0.40
	Female	1.66 ± 0.47	1.70 ± 0.59	2.01 ± 0.60	1.74 ± 0.54	1.95 ± 0.39
Potassium, mmol/L						
15 days	Male	6.51 ± 0.08	6.53 ± 0.12	6.49 ± 0.06	6.57 ± 0.10	6.52 ± 0.09
	Female	6.48 ± 0.09	6.50 ± 0.11	6.45 ± 0.11	6.51 ± 0.11	6.47 ± 0.10
Calcium, mmol/L						
15 days	Male	2.29 ± 0.07	2.27 ± 0.08	2.31 ± 0.07	2.32 ± 0.07	2.33 ± 0.06
	Female	2.31 ± 0.05	2.33 ± 0.10	2.33 ± 0.07	2.28 ± 0.08	2.30 ± 0.07
Sodium, mmol/L						
15 days	Male	139.34 ± 0.45	139.57 ± 0.59	139.10 ± 0.49	138.86 ± 0.61	139.45 ± 0.62
	Female	138.91 ± 0.59	139.16 ± 0.61	139.02 ± 0.61	139.27 ± 0.40	137.74 ± 0.53

**Table 7 cells-14-00341-t007:** Effect of a single intravenous WJ-MSC-SE33 administration on urinalysis in mice (M ± SEM).

Observation Period	Sex	Control (0.9% NaCl) (n = 12)	Control (WJ-MSC 2.5 × 10^7^ Cells/kg) (n = 12)	WJ-MSC-SE33		
				0.5 × 10^7^ Cells/kg (n = 12)	1.25 × 10^7^ Cells/kg (n = 12)	2.5 × 10^7^ Cells/kg (n = 12)
рН						
15 days	Male	6.33 ± 0.17	6.42 ± 0.15	6.33 ± 0.17	6.42 ± 0.15	6.42 ± 0.15
	Female	6.58 ± 0.15	6.50 ± 0.18	6.58 ± 0.15	6.42 ± 0.15	6.33 ± 0.11
Specific gravity, g/ml						
15 days	Male	1.003 ± 0.002	1.003 ± 0.002	1.003 ± 0.002	1.003 ± 0.002	1.004 ± 0.002
	Female	1.003 ± 0.002	1.004 ± 0.002	1.004 ± 0.002	1.002 ± 0.002	1.004 ± 0.002
Leukocytes, number per field						
15 days	Male	0-0	0-1	0-1	0-1	0-1
	Female	0-0	1-1	1-0	0-1	0-0
Erythrocytes, number per field						
15 days	Male	0-0	1-1	0-1	1-1	1-0
	Female	0-0	0-0	0-0	0-0	0-0
Glucose, mmol/L						
15 days	Male	negative	negative	negative	negative	negative
	Female	negative	negative	negative	negative	negative
Ketone bodies, mmol/L						
15 days	Male	negative	negative	negative	negative	negative
	Female	negative	negative	negative	negative	negative
Bilirubin, µmol/L						
15 days	Male	negative	negative	negative	negative	negative
	Female	negative	negative	negative	negative	negative
Total protein, g/L						
15 days	Male	0.00 ± 0.00	0.05 ± 0.05	0.00 ± 0.00	0.00 ± 0.00	0.00 ± 0.00
	Female	0.00 ± 0.00	0.05 ± 0.05	0.05 ± 0.05	0.00 ± 0.00	0.00 ± 0.00

**Table 8 cells-14-00341-t008:** Effect of a single intravenous WJ-MSC-SE33 administration on organ mass coefficients in mice (M ± SEM).

Observation Period	Sex	Control (0.9% NaCl) (n = 12)	Control (WJ-MSC 2.5 × 10^7^ Cells/kg) (n = 12)	WJ-MSC-SE33		
				0.5 × 10^7^ Cells/kg (n = 12)	1.25 × 10^7^ Cells/kg (n = 12)	2.5 × 10^7^ Cells/kg (n = 12)
Heart						
15 days	Male	4.94 ± 0.18	5.04 ± 0.18	4.93 ± 0.17	4.99 ± 0.11	5.06 ± 0.14
	Female	4.97 ± 0.19	4.86 ± 0.16	5.06 ± 0.19	4.95 ± 0.15	4.79 ± 0.14
Lungs (both)						
15 days	Male	9.80 ± 0.17	9.54 ± 0.35	9.42 ± 0.30	9.51 ± 0.37	9.73 ± 0.32
	Female	9.62 ± 0.31	9.48 ± 0.26	9.38 ± 0.26	9.60 ± 0.18	9.78 ± 0.30
Thymus						
15 days	Male	3.62 ± 0.08	3.58 ± 0.06	3.66 ± 0.08	3.60 ± 0.09	3.62 ± 0.08
	Female	3.73 ± 0.11	3.69 ± 0.09	3.70 ± 0.07	3.70 ± 0.09	3.60 ± 0.09
Spleen						
15 days	Male	6.29 ± 0.10	6.30 ± 0.17	6.06 ± 0.16	6.24 ± 0.11	6.24 ± 0.10
	Female	6.33 ± 0.14	6.26 ± 0.13	6.30 ± 0.12	6.47 ± 0.11	6.21 ± 0.11
Liver						
15 days	Male	58.50 ± 0.73	57.58 ± 1.05	59.07 ± 0.72	58.22 ± 1.02	59.16 ± 0.59
	Female	59.25 ± 1.53	59.64 ± 0.98	59.26 ± 1.00	59.10 ± 1.09	59.63 ± 1.02
Kidneys (both)						
15 days	Male	12.35 ± 0.32	12.57 ± 0.26	12.65 ± 0.26	12.30 ± 0.27	12.17 ± 0.25
	Female	12.40 ± 0.28	12.37 ± 0.27	12.48 ± 0.27	12.64 ± 0.39	12.29 ± 0.25
Brain						
15 days	Male	20.57 ± 0.34	20.75 ± 0.35	20.78 ± 0.29	20.40 ± 0.24	20.35 ± 0.14
	Female	20.67 ± 0.30	21.00 ± 0.26	20.81 ± 0.27	20.94 ± 0.32	20.61 ± 0.23

**Table 9 cells-14-00341-t009:** General anaphylactic reaction (*I_w_*) in animals following endolaryngeal and intravenous administration of WJ-MSC-SE33 at sensitizing doses (M ± SEM).

Negative Control (0.9% NaCl)		Positive Control (0.6% HEW)		WJ-MSC-SE33			
				0.5 × 10^7^ Cells/kg (*n* = 6)		2.5 × 10^7^ Cells/kg (*n* = 6)	
Male (n = 6)	Female (n = 6)	Male (n = 6)	Female (n = 6)	Male (n = 6)	Female (n = 6)	Male (n = 6)	Female (n = 6)
Endolaryngeal administration							
0	0	3.33 ± 0.33 *	3.50 ± 0.34 *	0	0	0	0
Intravenous administration							
0	0	3.00 ± 0.37 *	3.17 ± 0.31 *	0	0	0	0

*—statistically significant differences compared to the control group (0.9% NaCl), Mann–Whitney *U*-test, *p* < 0.05.

**Table 10 cells-14-00341-t010:** Semi-quantitative analysis of histological sections of mouse lungs after intravenous administration of WJ-MSC-SE33 expressing SE-33 peptide (M ± SEM).

Observation Period	Control (0.9% NaCl)	Control (Native WJ-MSCs at 0.5 × 10^7^ Cells/kg)	WJ-MSC-SE33		
			0.5 × 10^7^ Cells/kg	1 × 10^7^ Cells/kg	1.5 × 10^7^ Cells/kg
Infiltration or aggregation of inflammatory cells in the airspace or vessel walls					
5 days	2.20 ± 0.13	0.90 ± 0.18 *	0.90 ± 0.18 *	0.80 ± 0.20 *	0.80 ± 0.25 *
8 days	2.50 ± 0.22	1.00 ± 0.19 *	0.88 ± 0.23 *	0.89 ± 0.20 *	0.89 ± 0.26 *
Interstitial congestion and hyaline membrane formation					
5 days	3.30 ± 0.15	1.50 ± 0.17 *	1.40 ± 0.16 *	1.40 ± 0.22 *	1.30 ± 0.21 *
8 days	3.50 ± 0.22	1.50 ± 0.19 *	1.38 ± 0.18 *	1.44 ± 0.18 *	1.33 ± 0.24 *
Hemorrhage					
5 days	1.00 ± 0.00	0.40 ± 0.16 *	0.30 ± 0.15 *	0.20 ± 0.13 *	0.30 ± 0.15 *
8 days	1.00 ± 0.00	0.38 ± 0.18 *	0.38 ± 0.18 *	0.33 ± 0.17 *	0.22 ± 0.15 *

*—statistically significant differences compared to the control group (0.9% NaCl), Mann–Whitney *U*-test, *p* < 0.05.

## Data Availability

The original contributions presented in this study are included in the article and Appendix A. Further inquiries can be directed to the corresponding author.

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
