# Peer review of "Preclinical Evaluation of the Safety, Toxicity and Efficacy of Genetically Modified Wharton’s Jelly Mesenchymal Stem/Stromal Cells Expressing the Antimicrobial Peptide SE-33"

_cells, 2025, doi:10.3390/cells14050341_

Round 1

Reviewer 1 Report

Comments and Suggestions for Authors

The authors describe the preclinical safety in mice of a genetically modified human MSC expressing an antimicrobial peptide SE-33. Although MSC and also some genetically modified SC have been used in clinical studies with no or few side effects, it is argued that the need for another preclinical study on this specific MSC is warranted because often data on the use of other genetically modified cells is not easily accessible and this study will increase the knowledge about their safety profile. The study is carefully conducted and provides relevant information, which is certainly interesting to the reader in terms of results and methodology. Nevertheless, there are a number of remarks and questions, which should be addressed:

1. The term "mesenchymal stem cell" is outdated as MSC are not considered true stem cells. This should be changed and the more common term "mesenchymal stroma cell" used.

2. The safety profile and perhaps efficacy data of the SE-33 peptide alone should be described, if known. If not known, this should be stated. Furthermore, the potential clinical application may be described and the hypothesis of efficacy. Ideally, a potentially relevant efficacy test (preclinical) shpould be discussed.

3. It is stated that SE-33 remains inside cells after transfection. What is the expected mode of action if the active substance remains inside cells? Are there data on the production of the peptide in MSC-SE33 over time (e.g. immediately after transfection, 14 days after transfection)? This would be relevant for both, assessing safety and efficacy. 

4. Similarly, there is no data on the destiny of the MSC-SE33 cells (or control cells) post injection. Has the presence of human cells in different organs, or the circulation assessed at different time points after injection? has it been assessed whether SE33 is present in blood or urine of the injected mice?

5. There seems to be no or only neglectable local /systemic inflammatory responses to the allogenic human cells in an immunocompetent host background, even after multiple injections (challenges). Could the authors comment on this?

6. It is unclear how many cells were transfected with the SE-33 plasmid. It is stated that 100.000 cells were plated for transfection per well. The cells were directly used for the animal studies after transfection, apparently without further expansion? It should be further clarified a) which passage of the cells was used for transfection, b) whether cells were expanded after transfection, and which passage after transfection was used for injection into animals (or the antimicrobial activity assay), c) whether only one MSC - line (from one donor) was used, or more biological replicates were used and d) whether the SC were immortalized and to what passage the cells were viable. 

The manuscript may be acceptable for publication if the authors provide an informative point by point reply.

Author Response

Comments 1: The authors describe the preclinical safety in mice of a genetically modified human MSC expressing an antimicrobial peptide SE-33. Although MSC and also some genetically modified SC have been used in clinical studies with no or few side effects, it is argued that the need for another preclinical study on this specific MSC is warranted because often data on the use of other genetically modified cells is not easily accessible and this study will increase the knowledge about their safety profile. The study is carefully conducted and provides relevant information, which is certainly interesting to the reader in terms of results and methodology. Nevertheless, there are a number of remarks and questions, which should be addressed:

The term "mesenchymal stem cell" is outdated as MSC are not considered true stem cells. This should be changed and the more common term "mesenchymal stroma cell" used.

Response 1: Thank you for this important observation. We agree with your suggestion and have revised the text.

Comments 2: The safety profile and perhaps efficacy data of the SE-33 peptide alone should be described, if known. If not known, this should be stated. Furthermore, the potential clinical application may be described and the hypothesis of efficacy. Ideally, a potentially relevant efficacy test (preclinical) shpould be discussed.

Response 2: Thank you for this insightful suggestion. We have addressed this comment by adding information in the Introduction about the previously evaluated efficacy and preclinical study of the SE-33 peptide, including its antibacterial and antifungal properties, as well as its safety profile in another in vivo model. Relevant reference has been cited accordingly. Additionally, we have discussed the potential clinical application of SE-33 in the context of antibiotic-resistant bacterial infections and its advantages over natural analogues like LL-37. Moreover, we have expanded the Results section to include new data on the antimicrobial efficacy of WJ-MSC-SE33 against bacterial pneumonia in the in vivo model, further demonstrating the therapeutic potential of SE-33.

Comments 3: It is stated that SE-33 remains inside cells after transfection. What is the expected mode of action if the active substance remains inside cells? Are there data on the production of the peptide in MSC-SE33 over time (e.g. immediately after transfection, 14 days after transfection)? This would be relevant for both, assessing safety and efficacy.

Response 3: Thank you for raising this important point. In our study, SE-33 detection and antimicrobial properties were assessed 24 hours after the resting period following transfection. This time point was chosen as the cells were subsequently prepared for in vivo administration. In response to your inquiry about the mechanism of action, we expanded the Discussion section (Page 29, Lines 959-975) to propose potential SE-33 release mechanisms.

Comments 4: Similarly, there is no data on the destiny of the MSC-SE33 cells (or control cells) post injection. Has the presence of human cells in different organs, or the circulation assessed at different time points after injection? has it been assessed whether SE33 is present in blood or urine of the injected mice?

Response 4: Thank you for your thoughtful comment. We acknowledge the importance of assessing the fate of MSC-SE33 cells and the SE-33 peptide post-injection for a comprehensive safety evaluation.

We indeed evaluated the circulation of SE-33 in the blood and its presence in different tissues (longs, liver, kidneys, spleen) following WJ-MSC-SE33 administration. Our data showed that SE-33 was detectable in the bloodstream within 4 hours post-injection and persisted for up to 48 hours, depending on the administered dose. The peptide was also detected in liver and lung tissues at 4 and 8 hours post-injection (single intravenous) and remained detectable for up to 48 hours. The concentration of SE-33 and its retention time correlated with the number of injected cells. However, these data are already included in another manuscript focused on pharmacokinetics and biodistribution and were therefore not included in the present one.

Regarding the fate of administered WJ-MSC-SE33 cells, we conducted histological assessments for up to 18 months as part of our tumorigenicity and oncogenicity evaluation. No evidence of ectopic engraftment or tumor formation was observed in any of the examined tissues. Nevertheless, since these studies were performed in immunocompetent mice without a positive control (e.g., highly tumorigenic cells or immunodeficient models), the results cannot be considered definitive in terms of long-term engraftment potential. As these findings fall outside the primary scope of the current study and were not conducted under conditions optimized for cell fate tracking, we have not included them in this manuscript.

We appreciate your valuable comment and have clarified these points in this response letter while maintaining the scope of our manuscript. We hope this explanation adequately addresses your concern.

Comments 5: There seems to be no or only neglectable local /systemic inflammatory responses to the allogenic human cells in an immunocompetent host background, even after multiple injections (challenges). Could the authors comment on this?

Response 5: We appreciate your observation and have addressed this concern by expanding the Discussion section (Page 28, Lines 923-940) to explain our choice of immunocompetent C57BL/6 mice as the model system.

Comments 6: It is unclear how many cells were transfected with the SE-33 plasmid. It is stated that 100.000 cells were plated for transfection per well. The cells were directly used for the animal studies after transfection, apparently without further expansion? It should be further clarified a) which passage of the cells was used for transfection, b) whether cells were expanded after transfection, and which passage after transfection was used for injection into animals (or the antimicrobial activity assay), c) whether only one MSC - line (from one donor) was used, or more biological replicates were used and d) whether the SC were immortalized and to what passage the cells were viable.

Response 6: Thank you for your detailed questions regarding this protocol. We have thoroughly addressed each point by including the following information in the Materials and Methods section (Pages 5-6, Lines 203-233).

Reviewer 2 Report

Comments and Suggestions for Authors

The topic is interesting. Experimental methods are described thoroughly, but significant issues are noted. 

The aim of the study and novelty are not clear.

Human MSCs are known to produce LL-37, so why are WJ MSCs used despite not expressing this anti-microbial molecule with the need for genetic modification?

https://pmc.ncbi.nlm.nih.gov/articles/PMC3293245/

Also, similar studies have proved the LL-37 anti-microbial effect on MSCs in mice models, e.g.

https://www.sciencedirect.com/science/article/pii/S1525001620302483

so, why this study is novel? And previous studies should be acknowledged.

The design of the work:

The in vivo work (mice model) should simultaneously show the antimicrobial effect and safety assessment of WJ-MSCs expressing SE-33. This can be done using a model of infection or sepsis, and it is essential as the therapeutic dose for using these specific MSCs has not been proven in vivo. Also, the MSC dose that might be safe in vivo is not necessarily the therapeutic (anti-microbial) dose.

The results:

Histology/organ changes can be quantified or semi-quantified, as showing sections might not be enough.

Author Response

Comments 1: The topic is interesting. Experimental methods are described thoroughly, but significant issues are noted.

The aim of the study and novelty are not clear.

Human MSCs are known to produce LL-37, so why are WJ MSCs used despite not expressing this anti-microbial molecule with the need for genetic modification?

https://pmc.ncbi.nlm.nih.gov/articles/PMC3293245/

Also, similar studies have proved the LL-37 anti-microbial effect on MSCs in mice models, e.g.

https://www.sciencedirect.com/science/article/pii/S1525001620302483

so, why this study is novel? And previous studies should be acknowledged.

Response 1: Thank you for pointing this out. We appreciate this comment and have addressed it by including detailed information in the Introduction and Discussion sections of our manuscript to clarify the novelty and aim of our study. Additionally, we have revised the Introduction and Discussion sections to acknowledge previous studies on LL-37 that you have mentioned.

Comments 2: The design of the work: The in vivo work (mice model) should simultaneously show the antimicrobial effect and safety assessment of WJ-MSCs expressing SE-33. This can be done using a model of infection or sepsis, and it is essential as the therapeutic dose for using these specific MSCs has not been proven in vivo. Also, the MSC dose that might be safe in vivo is not necessarily the therapeutic (anti-microbial) dose.

Response 2: Thank you so much for your suggestion. We appreciate this valuable comment. To address this, we included additional data on the in vivo antimicrobial efficacy of WJ-MSC-SE33 using a bacterial lung infection model induced by S. aureus. The revised sections (Results and Discussion) provide comprehensive details (pages 24-29).

Comments 3: The results: Histology/organ changes can be quantified or semi-quantified, as showing sections might not be enough.

Response 3: Thank you for this important observation. We did not perform quantitative or semi-quantitative analysis of histological sections in the general toxicity studies because no significant differences were observed between the tissue samples from mice treated with WJ-MSC-SE33, native WJ-MSCs, and the control group. Therefore, we considered it sufficient to provide descriptive histopathological evaluations. This approach is consistent with other published studies (e.g., DOI: 10.1016/j.reth.2022.01.008), where qualitative descriptions are deemed adequate in the absence of pathological changes.

However, we conducted semi-quantitative analysis for the histological examination of lung tissues from mice with S. aureus-induced bacterial lung infection. The corresponding data were added to the Results section as Table 10 (Lines 851-854).

Round 2

Reviewer 1 Report

Comments and Suggestions for Authors

The authors have diligently and substantially responded to the questions and comments from the previous review. The manuscript is now appropriate for publicatrion.

Author Response

Comments 1: The authors have diligently and substantially responded to the questions and comments from the previous review. The manuscript is now appropriate for publicatrion.

Response 1: Thank you for your valuable contribution. We sincerely appreciate your thorough review of our manuscript and your constructive comments, which have greatly contributed to its improvement. Your feedback has helped us present our findings more clearly.

Reviewer 2 Report

Comments and Suggestions for Authors

The comments are mostly addressed. 

Still, the title, and other parts should highlight the novel aspects of the study: WJ-MSCs (other studies did bone marrow), staph pneumonia (other studies did sepsis or E-Coli pneumonia) and efficacy and safety study.

Author Response

Comments 1: The comments are mostly addressed. Still, the title, and other parts should highlight the novel aspects of the study: WJ-MSCs (other studies did bone marrow), staph pneumonia (other studies did sepsis or E-Coli pneumonia) and efficacy and safety study.

Response 1: We appreciate this valuable suggestion. To emphasize the novelty of our study, we have revised the title to highlight the use of WJ-MSCs and the combined assessment of efficacy and safety. Additionally, we have made further clarifications in the Results (Lines 782-789) and Discussion (Lines 946-957) sections. These revisions should now better reflect the aspects of our research.
